# PROCEEDINGS A

wave motion, acoustics, statistical physics

wave propagation, random media, backscattering, multiple scattering, ensemble averaging, Wiener–Hopf

**Author for correspondence:**
Artur L. Gower
e-mail: arturgower@gmail.com

# A proof that multiple waves propagate in ensemble-averaged particulate materials

Artur L. Gower[1], I. David Abrahams[2] and William J. Parnell[3]

[1]Department of Mechanical Engineering, University of Sheffield, Sheffield, UK
[2]Isaac Newton Institute for Mathematical Sciences, 20 Clarkson Road, Cambridge CB3 0EH, UK
[3]School of Mathematics, University of Manchester, Oxford Road, Manchester M13 9PL, UK

ALG, 0000-0002-3229-5451; WJP, 0000-0002-3676-9466

Effective medium theory aims to describe a complex inhomogeneous material in terms of a few important macroscopic parameters. To characterize wave propagation through an inhomogeneous material, the most crucial parameter is the *effective wavenumber*. For this reason, there are many published studies on how to calculate a single effective wavenumber. Here, we present a proof that there *does not* exist a unique effective wavenumber; instead, there are an infinite number of such (complex) wavenumbers. We show that in most parameter regimes only a small number of these effective wavenumbers make a significant contribution to the wave field. However, to accurately calculate the reflection and transmission coefficients, a large number of the (highly attenuating) effective waves is required. For clarity, we present results for scalar (acoustic) waves for a two-dimensional material filled (over a half-space) with randomly distributed circular cylindrical inclusions. We calculate the effective medium by ensemble averaging over all possible inhomogeneities. The proof is based on the application of the Wiener–Hopf technique and makes no assumption on the wavelength, particle boundary conditions/size or volume fraction.

This technique provides a simple formula for the reflection coefficient, which can be explicitly evaluated for monopole scatterers. We compare results with an alternative numerical matching method.

## 1. Introduction

Materials comprising particles or inclusions that are randomly distributed inside a uniform host medium occur frequently in the world around us. They occur as synthetically fabricated media and also in nature. Common examples include composites, emulsions, suspensions, complex gases and polymers. Understanding how electromagnetic, elastic or acoustic waves propagate through these materials is necessary in order to characterize the properties of these materials, and also to design new materials that can control wave propagation.

The wave scattered from a particulate material will be influenced by the positions and properties of all particles, which are usually unknown. However, this scattered field, averaged over space or over time, depends only on the average particle properties. Many measurement systems perform averaging over space, if the receivers or incident wavelength are large enough [1], or over time [2]. In most cases, this averaging process is the same as averaging over all possible particle configurations. Such systems are sometimes called ergodic [2,3]. In this paper, we focus on ensemble-averaged waves, satisfying the scalar wave equation in two dimensions, reflecting from, and propagating in, a half-space particulate material. In certain scenarios, such as light scattering, it is easier to measure the average intensity of the wave. However, even in these cases, the ensemble-averaged field is often needed as a first step [4,5].

One driving principle, often used in the literature, is that the ensemble-averaged wave itself satisfies a wave equation with a single effective wavenumber [6–8]. Reducing an inhomogeneous material, with many unknowns, down to one effective wavenumber is attractive as it greatly reduces the complexity of the problem. For this reason, many papers have attempted to deduce this unique effective wavenumber from first principles in electromagnetism [3,9,10], acoustics [11–15] and elasticity [16,17]. See [18] for a short overview of the history of this topic, including typical statistical assumptions employed within the methods, such as hole-correction and the quasi-crystalline approximation, which we also adopt here.

The assumption that the ensemble-averaged wave field satisfies a wave equation, with an effective wavenumber, has never been fully justified. Here we prove that there *does not* exist a unique effective wavenumber but instead there are an infinite number of them. Gower *et al.* [18] first showed that there exist many effective wavenumbers, and provided a technique, the *Matching Method*, to efficiently calculate the effective wave field. In the present paper and [18], we show that for some parameter regimes, at least two effective wavenumbers are needed to obtain accurate results, when compared with numerical simulations. We also provide examples of how a single effective wave approximation leads to inaccurate results for both transmission and reflection for a half-space filled with particles (figure 1).

Although the *Matching Method* developed in [18] gave accurate results, when compared to numerical methods and known asymptotic limits, the limitations of the method were not immediately clear. Here, however, we illustrate that the Matching Method is robust, because combining many effective wavenumbers is not just a good approximation, it is an analytical solution to the integral equation governing the ensemble-averaged wave field. We prove this by employing the Wiener–Hopf technique and then, for clarity, illustrate the solution for particles that scatter only in their monopole mode. The Wiener–Hopf technique also gives a simple and elegant expression for the reflection coefficient.

The Wiener–Hopf technique is a powerful tool to solve a diverse range of wave scattering problems, see [19, (ch. 5. Wiener–Hopf Technique)] and [20,21] for an introduction. It is especially useful for semi-infinite domains [22–27] and boundary value problems of mixed type. In this work, the Wiener–Hopf technique clearly reveals the form of the analytic solution, but to compute

**3**

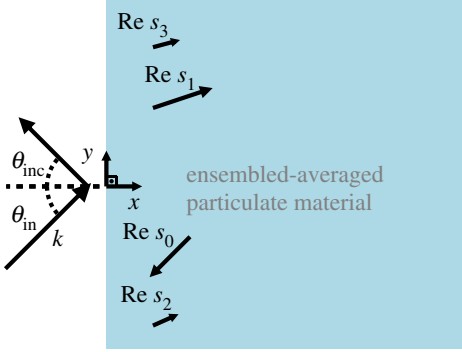

**Figure 1.** When an incident plane wave $e^{i\mathbf{k}\cdot(x,y)}$, with $\mathbf{k} = k(\cos\theta_{inc}, \sin\theta_{inc})$, encounters an (ensemble-averaged) particulate material, it excites many transmitted plane waves and one reflected plane wave. The transmitted waves are of the form $e^{i\mathbf{s}_p\cdot(x,y)}$ with wavenumbers $\mathbf{s}_p = S_p(\cos\theta_p, \sin\theta_p)$, where both $S_p$ and $\theta_p$ are complex numbers. The larger Im $s_p$, the more quickly the wave attenuates as it propagates into the half-space and the smaller the drawn vector for that wave above. The results shown here represent the effective wavenumbers for parameters (5.2), which are shown in figure 3. (Online version in colour.)

the solution would require an analytic factorization of a matrix function. To explicitly perform this factorization is difficult [28–31]. Indeed, this is often the hardest aspect of employing the Wiener–Hopf technique, although there exist approximate methods for this purpose [28,32–34]. We do not focus in this article on these analytic factorizations, as there already exists a method to compute the required solution [18]. Instead, the present work acts as proof that the Matching Method [18] faithfully reproduces the form of the analytic solution.

Figure 1 shows the main set-up and result of this paper: an incident plane wave excites the half-space $x > 0$ filled with ensemble-averaged particles (the blue region), which generates a reflected wave and many effective transmitted waves. The $s_p$ are the transmitted wavevectors, and the smaller the length of the vector, the faster that effective wave attenuates as it propagates further into the material.

The paper begins by summarizing the equations that govern ensemble-averaged waves in two dimensions in §2. Following this, in §3 we apply the Wiener–Hopf technique to the governing integral equation and deduce that the solution is a superposition of plane waves, each with a different effective wavenumber. A simple expression for the reflection coefficient is also derived. In §4, we specialize the results for particles that scatter only in the monopole mode, which leads to a closed form analytic solution.

The dispersion relation (3.30), derived in §3, admits an infinite number of solutions, the effective wavenumbers. In §5, we deduce asymptotic forms for the effective wavenumbers in both a low- and high-frequency limit. In §6, we compare numerical results for monopole scatterers, using the Wiener–Hopf technique, with classical methods that assume only one effective wavenumber [11,13], and the Matching Method introduced in [18]. In general, when comparing predicted reflection coefficients, the Wiener–Hopf and Matching Method agree well, whereas the classical single-effective-wavenumber method can disagree by anywhere up to 20%. These results are discussed in §7 together with anticipated future steps.

## 2. Waves in ensemble-averaged particles

Consider a region filled with particles or inclusions that are uniformly distributed. The field $u$ is governed by the scalar wave equations:

$$\nabla^2 u + k^2 u = 0, \quad \text{(in the background material)} \tag{2.1}$$

and

$$\nabla^2 u + k_o^2 u = 0, \quad \text{(inside a particle)}, \tag{2.2}$$

where $k$ and $k_o$ are the real wavenumbers of the background and inclusion materials, respectively. We assume all particles are identical, except for their position and orientation, for simplicity. For a distribution of particles, or multi-species, see [15].

Our goal is to calculate the ensemble-averaged field $\langle u(x, y) \rangle$, that is, the field averaged over all possible particle positions and orientations. For clarity, and ease of exposition, we consider that the particles are equally likely to be located anywhere except that they cannot overlap (this is often called the *hole correction* assumption). We also assume the quasi-crystalline approximation; for details on this, and for further details on deducing the results in this section, see [11,15,18].

By splitting the total steady wave field $u(x, y)$ into a sum of the incident wave $u_{\text{inc}}(x, y)$ and waves scattered by each particle, the $j$th scattered wave being $u_j(x, y)$, we can write

$$u(x, y) = u_{\text{inc}}(x, y) + \sum_j u_j(x, y). \tag{2.3}$$

A simple and useful scenario to consider is when all particles are placed only within the half-space[1] $x > 0$, which are then excited by a plane wave, with implicit time dependence $e^{i\omega t}$, incident from a homogeneous region:

$$u_{\text{inc}}(x, y) = e^{i(\alpha x + \beta y)}, \quad \text{with } (\alpha, \beta) = (k \cos \theta_{\text{inc}}, k \sin \theta_{\text{inc}}), \tag{2.4}$$

where we restrict the incident angle $-(\pi/2) < \theta_{\text{inc}} < (\pi/2)$, as shown in figure 1, and consider a slightly dissipative medium with

$$\text{Re}\, k > 0 \quad \text{and} \quad \text{Im}\, k > 0. \tag{2.5}$$

This dissipation will facilitate the use of the Wiener–Hopf technique, and after reaching the solution we can take $k$ to be real.[2]

To describe the particulate medium, we employ the following notation:

$$b = \text{the minimum distance between particle centres}, \tag{2.6}$$

$$\mathfrak{n} = \text{number of particles per unit area}, \tag{2.7}$$

$$T_n = \text{the coefficients of the particle's T-matrix}, \tag{2.8}$$

and

$$\phi = \frac{\pi \mathfrak{n} b^2}{4} = \text{particle area fraction}. \tag{2.9}$$

Although the area fraction $\phi$, normally called the volume fraction, is a combination of other parameters, it is useful because it is non-dimensional. If we let $a_o$ be the maximum distance from the particle's centre to its boundary, then we can set $b = \gamma a_o$, where $\gamma \geq 2$ so as to avoid two particles overlapping. The volume fraction that does not include the exclusion zone $\phi'$, as used in [18, (eqn (4.7))], is then $\phi' = 4\phi/\gamma^2$.

The $T_n$ are the coefficients of a diagonal T-matrix [35–39]. The T-matrix determines how the particle scatters waves, and so depends on the particle's shape and boundary conditions. A diagonal T-matrix can be used to represent either a radially symmetric particle, or particles averaged over their orientation, assuming the orientations have a random uniform distribution.

The results of ensemble averaging (2.3) from first principals are deduced in a number of references[15,18] and so details of this procedure are omitted here for brevity. To represent the

---

[1]The case where particles can be placed anywhere in the plane can lead to ill-defined integrals [11].

[2]Assuming $\text{Im}\, k = \epsilon > 0$, rather than $\epsilon \geq 0$, will facilitate calculating certain integrals that appear below. However, after reaching a solution, we can take the limit $\epsilon \to 0$ to recover the physically viable solution for $\text{Im}\, k = 0$.

ensemble-averaged scattered wave from a particle, whose centre is fixed at $(x_1, y_1)$, we use

$$\langle u_1(x_1 + X, y_1 + Y)\rangle_{(x_1, y_1)} = \sum_{n=-\infty}^{\infty} A_n(x_1) e^{i\beta y_1} H_n^{(1)}(kR) e^{in\Theta}, \qquad (2.10)$$

for $R := \sqrt{X^2 + Y^2} > b/2$, so that $(X, Y)$ is on the outside of this particle, with $(R, \Theta)$ being the polar coordinates of $(X, Y)$, $H_n^{(1)}$ are Hankel functions of the first kind, and $A_n$ is some field we want to determine.[3]

By choosing[4] $x < -b$, which is outside of the region filled with particles, then taking the ensemble average on both sides of (2.3) results in eqn (6.7) of [18], given by

$$\langle u(x, y)\rangle = u_{\text{inc}}(x, y) + \mathfrak{R} e^{-i\alpha x + i\beta y} \quad \text{for } x < -b, \qquad (2.11)$$

which is the incident wave plus an effective reflected wave with reflection coefficient

$$\mathfrak{R} = e^{i\alpha x} \mathfrak{n} \sum_{n=-\infty}^{\infty} \int_0^{\infty} A_n(x_1) \psi_n(x_1 - x) \, dx_1, \qquad (2.12)$$

where we assumed particles are distributed according to a uniform distribution, and the kernel $\psi_n$ is given by

$$\psi_n(X) = \int_{Y^2 > b^2 - X^2} e^{i\beta Y} (-1)^n H_n^{(1)}(kR) e^{in\Theta} \, dy. \qquad (2.13)$$

Later we show that, as expected, $\mathfrak{R}$ is independent of $x$.

The system governing $A_m(x)$ is given by eqn (4.7) of [18]:

$$\mathfrak{n} T_m \sum_{n=-\infty}^{\infty} \int_0^{\infty} A_n(x_2) \psi_{n-m}(x_2 - x_1) \, dx_2$$

$$= A_m(x_1) - e^{i\alpha x_1} T_m e^{im(\pi/2 - \theta_{\text{inc}})}, \quad \text{for } x_1 \geq 0, \quad (2.14)$$

for all integers $m$. Kristensson [40, (eqn 15)] presents an equivalent integral equation for electromagnetism and particles in a slab.

Our main aim is to reach an exact solution for $A_n(x)$ by employing the Wiener–Hopf technique to (2.14). We show how this also leads to simple solutions for the reflection coefficient by using (2.11). We acknowledge the authors of [11], as they noticed that (2.14) is a Wiener–Hopf integral equation, but apparently did not follow the steps indicated in the following sections.[5]

## 3. Applying the Wiener–Hopf technique

Equation (2.14) is convolution integral equation with a difference kernel. This means applying a Fourier transform can lead to elegant and simple solutions. To facilitate, we must analytically extend (2.14) for all $x_1 \in \mathbb{R}$ by defining

$$\mathfrak{n} T_m \sum_{n=-\infty}^{\infty} \int_0^{\infty} A_n(x_2) \psi_{n-m}(x_2 - x_1) \, dx_2$$

$$= \begin{cases} A_m(x_1) - e^{i\alpha x_1} T_m e^{im(\pi/2 - \theta_{\text{inc}})}, & x_1 \geq 0, \\ D_m(x_1), & x_1 < 0, \end{cases} \qquad (3.1)$$

[3]The factor $e^{i\beta y_1}$ appears due to the translational invariance of $\langle u_1(x_1, y_1)\rangle$ in $y_1$, which is a result of the material being statistically homogeneous, see [15] for details.

[4]We define the reflection coefficient only for $x < -b$, instead of $x < -b/2$, so that we can use $\psi_n$ in the formula for $\mathfrak{R}$, which will in turn facilitate calculating $\mathfrak{R}$.

[5]However, they were unable to solve it because, it seems, of a mistake in the integrand of eqn (37) of [11]; they used $e^{i\beta Y}$ where they should have used $\cos(\beta Y)$.

for integers $m$, where if the $A_n(x)$ were known for $x > 0$, then the $D_n(x)$ would be given from the left-hand side. Note that the kernel $\psi_n$ defined in (2.13) is already analytic in the domain $\mathbb{R}$.

The field $D_0(x)$ is not just an abstract construct, it is closely related to the reflected wave: by directly comparing (3.1) with the reflection coefficient (2.12), for $x < -b$, we find that

$$D_0(x) = T_0 \mathfrak{Re}^{-i\alpha x}. \tag{3.2}$$

To solve (3.1), we employ the Fourier transform and its inverse, which we define as

$$\hat{f}(s) = \int_{-\infty}^{\infty} f(x) e^{isx} \, dx \quad \text{with} \quad f(x) = \frac{1}{2\pi} \int_{-\infty}^{\infty} \hat{f}(s) e^{-isx} \, dx, \tag{3.3}$$

for any smooth function $f$. We then define

$$\hat{A}_n^+(s) = \int_0^{\infty} A_n(x) e^{isx} \, dx \quad \text{and} \quad \hat{D}_n^-(s) = \int_{-\infty}^0 D_n(x) e^{isx} \, dx. \tag{3.4}$$

We can determine where $\hat{A}_n^+$ and $\hat{D}_n^-$ are analytic by assuming[6] that

$$|A_n(x)| < e^{-xc} \quad \text{for } x \to \infty \tag{3.5}$$

and

$$|D_n(x)| < e^{xc} \quad \text{for } x \to -\infty, \tag{3.6}$$

for some (possibly small) positive constant $c$. This leads to $\hat{A}_n^+(s)$ being analytic for $\mathrm{Im}\, s > -c$, while $\hat{D}_n^-(s)$ is analytic for $\mathrm{Im}\, s < c$. In other words, both $\hat{A}_n^+(s)$ and $\hat{D}_n^-(s)$ are analytic in the overlapping strip

$$|\mathrm{Im}\, s| < c. \tag{3.7}$$

To apply the Wiener–Hopf technique, we also need to specify the large $s$ behaviour for both $\hat{A}_n^+(s)$ and $\hat{D}_n^+(s)$. To achieve this, we assume, on physical grounds, that $A_n(x)$ is bounded when $x \to 0^+$, and $D_n(x)$ is bounded when $x \to 0^-$. Then, it can be shown [21,41] that

$$\hat{A}_n^+(s) = \mathcal{O}(|s|^{-1}) \quad \text{and} \quad \hat{D}_n^-(s) = \mathcal{O}(|s|^{-1}) \quad \text{for } |s| \to \infty, \tag{3.8}$$

in their respective half-planes of analyticity.

Applying a Fourier transform to both sides of equation (3.1), the left-hand side becomes

$$\int_0^{\infty} A_n(x_2) \int_{-\infty}^{\infty} \psi_{n-m}(x_1 - x_2) e^{isx_1} \, dx_1 \, dx_2 = \hat{A}_n^+(s) \hat{\psi}_{n-m}(s), \tag{3.9}$$

in which $\hat{\psi}_n(s)$ is well defined (i.e. analytic) for $s$ in the strip:

$$|\mathrm{Im}\, s| < (1 - |\sin \theta_{\text{inc}}|) \mathrm{Im}\, k, \tag{3.10}$$

see appendix A for details. The right-hand side of (3.1) becomes

$$\int_{-\infty}^0 D_m(x_1) e^{isx_1} \, dx_1 + \int_0^{\infty} A_m(x_1) e^{isx_1} \, dx_1$$
$$- e^{im(\pi/2 - \theta_{\text{inc}})} T_m \int_0^{\infty} e^{ix_1(s+\alpha)} \, dx_1 = \hat{D}_m^-(s) + \hat{A}_m^+(s) - T_m \frac{i e^{im(\pi/2 - \theta_{\text{inc}})}}{(s+\alpha)^+}, \tag{3.11}$$

where for the last step we assumed $\mathrm{Im}\,(s + \alpha) > 0$, which is why we use the superscript $+$ on $(s + \alpha)^+$. This assumption, together with (3.7) and (3.10), is satisfied if

$$|\mathrm{Im}\, s| < \epsilon, \quad \text{where } \epsilon = \min\{c, (1 - |\sin \theta_{\text{inc}}|) \mathrm{Im}\, k, \mathrm{Im}\, \alpha\}. \tag{3.12}$$

If (3.12) is satisfied, then we can combine (3.9), (3.11) and (A 6), to obtain the Fourier transform of (3.1) in matrix form:

$$\frac{\boldsymbol{\Psi}(s) \hat{\boldsymbol{A}}^+(s)}{s^2 - \alpha^2} = -\hat{\boldsymbol{D}}^-(s) + \frac{\boldsymbol{B}}{s + \alpha}, \tag{3.13}$$

[6]The solutions for $A_n(x)$ and $D_n(x)$, in the next section, show that these assumptions do hold.

where $\hat{A}^+(s)$ and $\hat{D}^-(s)$ are vectors with components $\hat{A}_n^+(s)$ and $\hat{D}_n^-(s)$, respectively, and

$$B_m = iT_m e^{im(\pi/2 - \theta_{\text{inc}})}, \tag{3.14}$$

$$\Psi_{mn}(s) = G_{mn}(S)(-i)^{n-m} e^{i(n-m)\theta_S}, \tag{3.15}$$

$$G_{mn}(S) = (s^2 - \alpha^2)\delta_{mn} + 2\pi \mathfrak{n} T_m N_{n-m}(bS) \tag{3.16}$$

and

$$N_m(bS) = bk J_m(bS)H_m^{(1)'}(bk) - bS J_m'(bS)H_m^{(1)}(bk) \tag{3.17}$$

where, for reference,

$$\Psi_{mn}(s) = (s^2 - \alpha^2)\left[\delta_{mn} - \mathfrak{n} T_m \hat{\psi}_{n-m}(s)\right], \tag{3.18}$$

and $\hat{\psi}_{n-m}(s)$ is given by (A 6). In the above, $\theta_S$ and $S$ are chosen to satisfy

$$s = S\cos\theta_S \quad \text{with} \quad S\sin\theta_S = k\sin\theta_{\text{inc}}. \tag{3.19}$$

Later we identify $S$ and $\theta_S$ as the effective wavenumber and transmission angle. The above does not determine the sign of $S$ for any given complex $s$. To fully determine $S$ and $\theta_S$, we take $\text{sgn}(\text{Re}\, s) = \text{sgn}(\text{Re}\, S)$ which together with (3.19) leads to

$$\theta_S = \arctan\left(\frac{k\sin\theta_{\text{inc}}}{s}\right) \quad \text{and} \quad S = \sqrt{s^2 + (k\sin\theta_{\text{inc}})^2}, \tag{3.20}$$

where both $S$ and $\theta_S$, when considered as functions[7] of $s$, contain branch-points at $s = \pm ik\sin\theta_{\text{inc}}$ with finite branch-cut running between $-ik\sin\theta_{\text{inc}}$ and $ik\sin\theta_{\text{inc}}$. However, $\boldsymbol{\Psi}(s)$ is an entire matrix function having only zeros in $s$ and no branch-points; see the end of appendix A for details.

Determining the roots of $\det\boldsymbol{\Psi}(s) = 0$ will be a key step in solving (3.13), and so the following identities will be useful:

$$\Psi_{mn}(-s)T_n = \Psi_{mn}(s)T_n(-1)^{m-n}e^{2i(m-n)\theta_s} = \Psi_{nm}(s)T_m \tag{3.21}$$

and

$$\det\boldsymbol{\Psi}(-s) = \det\boldsymbol{\Psi}(s) \quad \text{and} \quad \det\boldsymbol{\Psi}(s) = \det\mathbf{G}(S). \tag{3.22}$$

where (3.21) results from (3.18) and (A 10). Equation (3.22) then follows from using (3.21)$_1$, (3.15) and appendix C.

## (a) Multiple waves solution

To solve (3.13), we use a matrix product factorization [42] of the form:

$$\boldsymbol{\Psi}(s) = \boldsymbol{\Psi}^-(s)\boldsymbol{\Psi}^+(s), \tag{3.23}$$

where $\boldsymbol{\Psi}^-(s)$, and its inverse, are analytic in $\text{Im}\, s < \epsilon$, and $\boldsymbol{\Psi}^+(s)$, and its inverse, are analytic for $\text{Im}\, s > -\epsilon$. See (3.12) for the definition of $\epsilon$.

For our purposes, it is enough to know that such a factorization exists [42], as this will lead to a proof that $A(x)$ is a sum of attenuating plane waves.

Multiplying both sides of (3.13) by $[\boldsymbol{\Psi}^-(s)]^{-1}$ and by $(s-\alpha)_-$ leads to

$$\frac{\boldsymbol{\Psi}^+(s)\hat{A}^+(s)}{(s+\alpha)_+} = -(s-\alpha)_-[\boldsymbol{\Psi}^-(s)]^{-1}\hat{D}^-(s) + [\boldsymbol{\Psi}^-(s)]^{-1}B\frac{(s-\alpha)_-}{(s+\alpha)_+}, \tag{3.24}$$

where $(s+\alpha)_+$ is analytic for $\text{Im}\, s > -\text{Im}\,\alpha$, while $(s-\alpha)_-$ is analytic for $\text{Im}\, s < \text{Im}\,\alpha$. We need to rewrite the last term above as a sum of a function which is analytic in the upper half-plane

---

[7]With our choice of branch-cut, the simplest way to compute $S$, with most software packages, is to use $S = \text{sgn}(\text{Re}\, s)\sqrt{s^2 + (k\sin\theta_{\text{inc}})^2}$ with the default cut location for the square root function, i.e. along the real negative line.

(Im $s > -\epsilon$) and another analytic in the lower half-plane. This is achieved below

$$
[\boldsymbol{\Psi}^-(s)]^{-1}\boldsymbol{B}\frac{(s-\alpha)_-}{(s+\alpha)_+} = -\underbrace{\frac{2\alpha}{(s+\alpha)_+}[\boldsymbol{\Psi}^-(-\alpha)]^{-1}\boldsymbol{B}}_{g^+(s)}
$$

$$
+ \underbrace{[\boldsymbol{\Psi}^-(s)]^{-1}\boldsymbol{B}\frac{(s-\alpha)_-}{(s+\alpha)_+} + [\boldsymbol{\Psi}^-(-\alpha)]^{-1}\boldsymbol{B}\frac{2\alpha}{(s+\alpha)_+}}_{g^-(s)}, \tag{3.25}
$$

where we define

$$
\lim_{s\to-\alpha} g^-(s) = \left[\boldsymbol{I} + 2\alpha[\boldsymbol{\Psi}^-(-\alpha)]^{-1}\frac{d\boldsymbol{\Psi}^-}{ds}(-\alpha)\right][\boldsymbol{\Psi}^-(-\alpha)]^{-1}\boldsymbol{B},
$$

so that $g^-(s)$ does not have a pole at $s = -\alpha$ and is therefore analytic for Im $s < \epsilon$.

Substituting (3.25) into (3.24) leads to

$$
\frac{\boldsymbol{\Psi}^+(s)\hat{\boldsymbol{A}}^+(s)}{(s+\alpha)_+} + g^+(s) = -(s-\alpha)_-[\boldsymbol{\Psi}^-(s)]^{-1}\hat{\boldsymbol{D}}^-(s) + g^-(s). \tag{3.26}
$$

Because both sides are analytic in the strip $|\text{Im } s| < \epsilon$, we can equate each side to $E(s)$, some analytic function in the strip. Further, as the left-hand side (right-hand side) of (3.26) is analytic for Im $s > \epsilon$ ( Im $s < -\epsilon$), we can analytically continue $E(s)$ for all $s$, i.e. $E(s)$ is entire.

To determine $E(s)$, we need to estimate its behaviour as $|s| \to \infty$. From (3.8), we have that $A^+(s) = (|s|^{-1})$ as $|s| \to \infty$ in the upper half-plane, and from (3.15) to (3.17):

$$
\boldsymbol{\Psi}(s) = (s^2 - \alpha^2)\boldsymbol{I} + \mathcal{O}(|s|) \quad \text{as } |s| \to \infty, \tag{3.27}
$$

for $s$ in the strip (3.12). From this, we know that the factors $\boldsymbol{\Psi}^+(s)$ and $\boldsymbol{\Psi}^-(s)$ must be $\mathcal{O}(|s|)$ as $|s| \to \infty$, in their respective half-planes of analyticity [28]. So, the left-hand side of (3.26) behaves as $\mathcal{O}(|s|^{-1})$ as $|s| \to \infty$ in Im $s > -\epsilon$. We can therefore use Liouville's theorem to conclude that $E(s) \equiv 0$, which means the Wiener–Hopf equation (3.26) is formally equivalent to

$$
\hat{\boldsymbol{A}}^+(s) = -2\alpha[\boldsymbol{\Psi}^+(s)]^{-1}[\boldsymbol{\Psi}^-(-\alpha)]^{-1}\boldsymbol{B} \tag{3.28}
$$

and

$$
\hat{\boldsymbol{D}}^-(s) = \frac{\boldsymbol{\Psi}^-(s)g^-(s)}{(s-\alpha)_-}. \tag{3.29}
$$

Let $\boldsymbol{C}^+(s)$ be the cofactor matrix of $\boldsymbol{\Psi}^+(s)$, so that

$$
[\boldsymbol{\Psi}^+(s)]^{-1} = \frac{[\boldsymbol{C}^+(s)]^{\mathrm{T}}}{\det(\boldsymbol{\Psi}^+(s))}.
$$

From the property (3.22)$_1$, we can write $\det \boldsymbol{\Psi}(s) = f(s^2)$ for some function $f$. Then, for every root $s = s_p$ of $\det \boldsymbol{\Psi}(s)$, with Im $s_p > 0$, we have that $-s_p$ is also a root, and vice-versa. From here onwards, we assume:

$$
\det \boldsymbol{\Psi}(s_p) = \det \boldsymbol{\Psi}(-s_p) = 0 \quad \text{with Im } s_p > 0 \quad \text{and} \quad p = 1, 2, \ldots, \infty. \tag{3.30}
$$

For any truncated matrix $\boldsymbol{\Psi}(s)$, i.e. evaluating $m, n = -M, \ldots, M$ in (3.15), the roots $s_p$ are discrete. In §5, we demonstrate asymptotically that they are indeed discrete for the limits of low and high wavenumber $k$. For the numerical results presented in this paper, we numerically solve the above dispersion relation for the truncating the matrix $\boldsymbol{\Psi}(s)$, and then increase $M$ until the roots converge (typically no more than $M = 4$ was required).

Given $\det \boldsymbol{\Psi}(s) = \det \boldsymbol{\Psi}^-(s) \det \boldsymbol{\Psi}^+(s)$, every root of $\det \boldsymbol{\Psi}(s)$ must either be a root of $\det \boldsymbol{\Psi}^-(s)$ or a root of $\det \boldsymbol{\Psi}^+(s)$. For $[\boldsymbol{\Psi}^+(s)]^{-1}$ to be analytic in the upper half-plane, $\det \boldsymbol{\Psi}^+(s)$ must only have roots $s = -s_p$. As a consequence, $\det \boldsymbol{\Psi}^-(s)$ only has roots $s = s_p$.

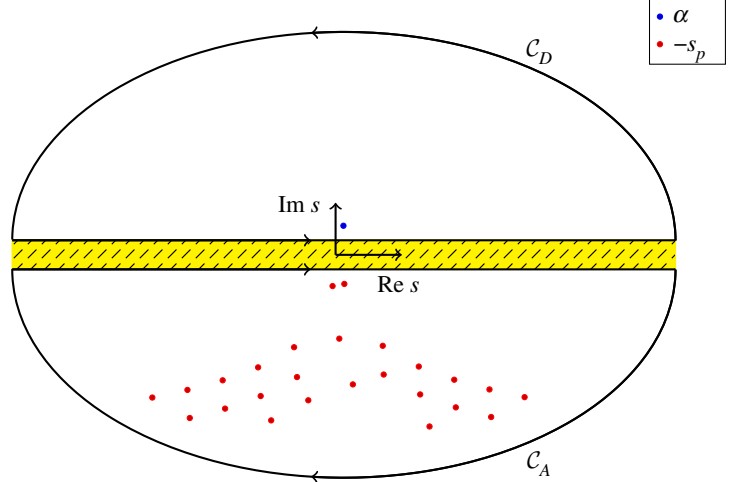

**Figure 2.** An illustration of the contour integral over $\mathcal{C}_D$, used to calculate (3.34) for $x < 0$, and the contour integral over $\mathcal{C}_A$, used to calculate (3.32) for $x > 0$. The $-s_p$ (the red points) are roots of (3.30), and also the poles of (3.28). The single blue point $\alpha$ is the only pole of (3.29). (Online version in colour.)

To use the residue theorem below, we need to calculate $\det \mathbf{\Psi}^+(s)$ for $s$ close to the root $-s_p$, in the form

$$\det \mathbf{\Psi}^+(s) = \det \mathbf{\Psi}^+(-s_p) + (s + s_p) \frac{\mathrm{d}\det \mathbf{\Psi}^+}{\mathrm{d}s}(-s_p) + \mathcal{O}((s + s_p)^2)$$

$$= \frac{s + s_p}{\det \mathbf{\Psi}^-(-s_p)} \frac{\mathrm{d}\det \mathbf{\Psi}}{\mathrm{d}s}(-s_p) + \mathcal{O}((s + s_p)^2), \tag{3.31}$$

where we use $(\mathrm{d}\det \mathbf{\Psi}/\mathrm{d}s)(-s_p)$ instead of $(\mathrm{d}\det \mathbf{\Psi}^+/\mathrm{d}s)(-s_p) \det \mathbf{\Psi}^-(-s_p)$, because it is more difficult to numerically evaluate $(\mathrm{d}\det \mathbf{\Psi}^+/\mathrm{d}s)(-s_p)$.

Using the above, and that $\mathbf{C}^+(S)$ is analytic for $\mathrm{Im}\, s > -\epsilon$, we can apply an inverse Fourier transform $(3.3)_2$ to both sides of (3.28) and using residue calculus we find

$$A(x) = -\frac{\alpha}{\pi} \int_{-\infty}^{\infty} \frac{[\mathbf{C}^+(s)]^{\mathrm{T}}[\mathbf{\Psi}^-(-\alpha)]^{-1}\mathbf{B}}{\det \mathbf{\Psi}^+(s)} \mathrm{e}^{-\mathrm{i}sx}\, \mathrm{d}s = \begin{cases} \sum_{p=1}^{\infty} A^p \mathrm{e}^{\mathrm{i}s_p x}, & x > 0, \\ 0, & x < 0, \end{cases} \tag{3.32}$$

with $\quad A^p = 2\alpha\mathrm{i} \frac{\det \mathbf{\Psi}^-(-s_p)}{(\mathrm{d}\det \mathbf{\Psi}/\mathrm{d}s)(-s_p)} [\mathbf{C}^+(-s_p)]^{\mathrm{T}}[\mathbf{\Psi}^-(-\alpha)]^{-1}\mathbf{B}. \tag{3.33}$

For $x > 0$, the integral over $s \in [-\infty, \infty]$ in (3.32) is, by Jordan's lemma, the same as a clockwise integral over the closed contour $\mathcal{C}_A$ which surrounds the poles $-s_1, -s_2, \ldots$, i.e. roots of (3.30), as shown by figure 2. Note that the cofactor matrix $\mathbf{C}^+(s)$ contains no poles and so does not contribute additional residual terms. The yellow striped region in figure 2 is the domain where $\mathbf{\Psi}$ is analytic. On the other hand, for $x < 0$, the integral (3.32) is the same as an integral over the counter-clockwise closed contour within the region $\mathrm{Im}\, s > 0$ (not shown in figure 2). The integrand has no poles in this domain and hence evaluates to zero.

Likewise, by applying an inverse Fourier transform to (3.29), we obtain

$$D(x) = \frac{1}{2\pi} \int_{-\infty}^{\infty} \frac{\mathbf{\Psi}^-(s)g^-(s)}{(s - \alpha)_-} \mathrm{e}^{-\mathrm{i}sx}\mathrm{d}s = \begin{cases} \mathrm{i}\mathbf{\Psi}^-(\alpha)[\mathbf{\Psi}^-(-\alpha)]^{-1}\mathbf{B}\mathrm{e}^{-\mathrm{i}\alpha x}, & x < 0, \\ 0, & x > 0. \end{cases} \tag{3.34}$$

For $x < 0$, the above integral is the same as a counter-clockwise closed integral over $\mathcal{C}_D$ which surrounds the pole $s = \alpha$ (recalling that $\mathrm{Im}\, \alpha > 0$), as shown in figure 2. The result is just the residue at this pole. That is, the function $\mathbf{\Psi}^-(s)g^-(s)$ contains no other singularities within $\mathrm{Im}\, s > 0$.

On the other hand, for $x > 0$ the integral is the same as a closed clockwise integral around the region $\text{Im } s < 0$ which evaluates to zero, as there are no singularities in this region (not shown in figure 2).

Clearly (3.32) shows that $A(x)$ is a sum of plane waves with different effective wavenumbers $s_p$, each satisfying (3.30). In §5, we discuss these roots in more detail, and in §4, we see that usually only a few effective wavenumbers are required to obtain accurate results.

## (b) Reflection coefficient

Substituting (3.34) in (3.2) leads to

$$\mathfrak{R} = i T_0^{-1} \sum_{n,m=-\infty}^{\infty} \Psi_{0n}^{-}(\alpha)[\mathbf{\Psi}^{-}(-\alpha)]_{nm}^{-1} B_m. \tag{3.35}$$

Alternatively, the reflection coefficient can be calculated from (2.12) by employing the form of $A(x)$ from (3.32), which is the more common approach. To simplify, we use

$$\psi_n(X) = (-1)^n \int_{-\infty}^{\infty} e^{ikY \sin\theta_{\text{inc}}} H_n^{(1)}(kR) e^{in\Theta} \, dY = \frac{2}{\alpha} i^n e^{-in\theta_{\text{inc}}} e^{i\alpha X} \quad \text{for } X > 0, \tag{3.36}$$

which then implies that $\psi_n(x_1 - x) = (2/\alpha)i^n e^{-in\theta_{\text{inc}}} e^{i\alpha(x_1 - x)}$ for $x_1 \geq x$. The above is shown in [43, (eqn (37))] and [11, (eqn (65))]. This result together with (3.32) substituted into (2.12) leads to the form

$$\mathfrak{R} = \frac{2\mathfrak{n}}{\alpha} \sum_{n=-\infty}^{\infty} i^n e^{-in\theta_{\text{inc}}} \int_0^{\infty} A_n(x_1) e^{i\alpha x_1} dx_1 = \frac{2i\mathfrak{n}}{\alpha} \sum_{n=-\infty}^{\infty} \sum_{p=1}^{\infty} i^n e^{-in\theta_{\text{inc}}} \frac{A_n^p}{s_p + \alpha}, \tag{3.37}$$

where we used that $\text{Im } s_p > 0$. The above agrees with [13, (eqn (39))] and[8] [18, (eqn (6.9))].

# 4. Monopole scatterers

For particles that scatter only in their monopole mode, i.e. the scattered waves are angularly symmetric about each particle, we can easily calculate the factorization (3.23). This type of scattered wave tends to dominate in the long wavelength limit for scatterers with Dirichlet boundary conditions. In acoustics, these correspond to particles with low density or low sound speed.

Once we know the factorization (3.23), we can then calculate the average scattering coefficient (3.32) and average reflection coefficient (3.35). We will compare both of these against predictions from other methods in §6.

## (a) Wiener–Hopf factorization

For scalar problems, there are well-known techniques to factorize $\Psi_{00}(s) = \Psi_{00}^{-}(s)\Psi_{00}^{+}(s)$, such as Cauchy's integral formulation, for details see [19, (Section 5. Wiener–Hopf Technique)] and [21].

For monopole scatterers we use $S^2 - k^2 = s^2 - \alpha^2$ and rewrite

$$\Psi_{00}(s) = (s^2 - \alpha^2)q(s), \quad \text{with } q(s) = 1 + 2\pi\mathfrak{n}\frac{T_0 N_0(bS)}{S^2 - k^2},$$

with $N_0(bS)$ given by (3.17). Then, because $q(s) \to 1$ as $|s| \to \infty$, we can factorize $q(s) = q^{-}(s)q^{+}(s)$ using

$$q^{+}(s) = \exp\left(\frac{1}{2\pi i} \fint_{-\infty}^{\infty} \frac{\log q(z)}{z - s} \, dz\right) \tag{4.1}$$

and

$$q^{-}(s) = \exp\left(-\frac{1}{2\pi i} \fint_{-\infty}^{\infty} \frac{\log q(z)}{z - s} \, dz\right), \tag{4.2}$$

[8]When taking a zero thickness boundary layer, i.e. $J = 0$, and appropriate substitutions.

where the integral path for $q^+(s)$ ($q^-(s)$) has to be in the strip where $q(s)$ is analytic, with the path for $q^+(s)$ ($q^-(s)$) passing below (above) $z$. We then have[9] that

$$\Psi_{00}^+(s) = (s+\alpha)_+ q^+(s), \quad \Psi_{00}^-(s) = (s-\alpha)_- q^-(s), \quad \Psi_{00}^-(-s) = -\Psi_{00}^+(s), \tag{4.3}$$

where $(4.3)_3$ holds if $-s$ is below the integration path of (4.2) and $s$ is above the integration path of (4.1). From (3.32), we see that we need only evaluate $\Psi_{00}^+(s)$, and therefore $q^+(s)$, for $s = s_1, s_2, \ldots, s_p$ where as $p$ increases, the $s_p$ become more distant from the real line. Then for large $z$, by inspection of (3.17), we have that

$$\left| \frac{\log q(z)}{z-s} \right| \sim \frac{1}{z^{3/2}} \frac{1}{|z-s|},$$

and therefore we can accurately approximate the integral (4.1) by truncating the integration domain for large $z$.

## (b) Explicit solution for monopole scatterers

For monopole scatterers $A_n(x) = D_n(x) = 0$ for $|n| > 0$. Using this in (3.14)–(3.17)) leads to all vectors and matrices having only one component, given by setting $n = m = 0$. In this case, $A$ (3.32) reduces to

$$A_0(x) = \sum_{p=1}^{\infty} A_0^p e^{is_p x} \quad \text{with} \quad A_0^p = \frac{2\alpha T_0}{\Psi_{00}^+(\alpha)} \frac{\Psi_{00}^+(s_p)}{(d\Psi_{00}/ds)(s_p)} = \frac{T_0}{s_p - \alpha} \frac{q^+(s_p)}{q^+(\alpha)q'(s_p)}, \tag{4.4}$$

for $x > 0$, where we used (4.3), $\mathbf{C}^+(s) = 1$, $\mathbf{B} = iT_0$, and $(d\Psi_{00})/(ds)(-s) = -(d\Psi_{00})/(ds)(s)$ for every $s$. Likewise for (3.35), we arrive at

$$\Re = \frac{\Psi_{00}(\alpha)}{(\Psi_{00}^+(\alpha))^2} = \frac{\pi n T_0 N_0(b\alpha)}{2(\alpha q^+(s))^2}. \tag{4.5}$$

Alternatively, using (3.37), we can calculate the contribution of $P$ effective waves to the reflection coefficient

$$\Re^P = \frac{2in}{\alpha} \sum_{p=1}^{P} \frac{A_0^p}{s_p + \alpha} = \frac{2in T_0}{\alpha q^+(\alpha)} \sum_{p=1}^{P} \frac{1}{s_p^2 - \alpha^2} \frac{q^+(s_p)}{q'(s_p)} \quad \text{with} \quad \Re = \lim_{P \to \infty} \Re^P, \tag{4.6}$$

where the error $|\Re^P - \Re|$ then indicates how many effective waves are needed to accurately describe the field near the boundary $x = 0$.

## 5. Multiple effective wavenumbers

Equation (3.32) clearly shows that $A(x)$ is a sum of attenuating plane waves, each with a different effective wavenumber $s_p$. These $s_p$ satisfy the dispersion equation (3.30):

$$\det \boldsymbol{\Psi}(s_p) = \det \mathbf{G}(S_p) = 0, \tag{5.1}$$

with $\boldsymbol{\Psi}$ given by (3.16) and the first identity follows from (3.22).

An important conclusion from $\det \mathbf{G}(S_p) = 0$ is that the wavenumbers $S_p$ are independent of the angle of incidence $\theta_{inc}$. We focus on showing the results for $S_p$, rather than $s_p$, because then we do not need to specify $\theta_{inc}$.

---

[9]Note that the factors $q^+(s)$ and $q^-(s)$ are singularity and pole free in their respective regions of analyticity, and so their inverses $[q^+(s)]^{-1}$ and $[q^-(s)]^{-1}$ have the same property.

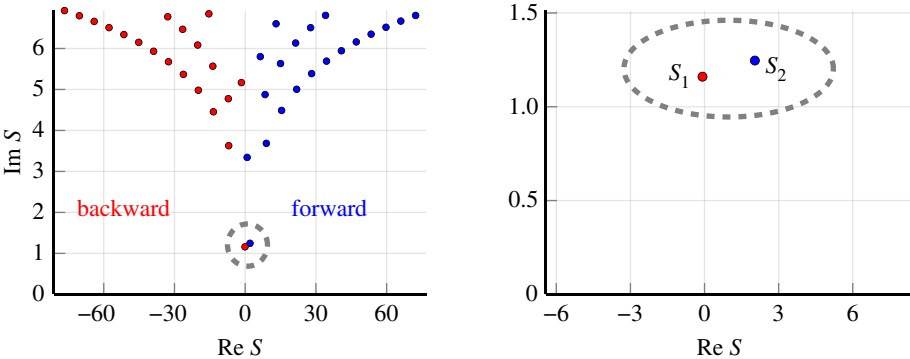

**Figure 3.** Examples of effective wavenumbers $S_p$ which satisfy the dispersion equation (5.1) with the properties (5.2). The blue points represent waves travelling forwards (i.e. deeper into the material), while the red represent waves travelling backwards. All these waves are excited in a reflection experiment. Two wavenumbers in particular stand out as having the lowest attenuation $S_1$ and $S_2$, both inside the grey dashed circle. The graph on the right is a magnification of the region close to these two wavenumbers. Out of these two, most efforts in the literature have focused on calculating $S_2$, as it often has the lowest imaginary part; however, for this case, because $S_1$ has a smaller attenuation it will have a significant contribution to both transmission and reflection. (Online version in colour.)

As a specific example, let us consider circular particles with Dirichlet boundary conditions (i.e. particles with zero density or soundspeed), and the parameters

$$T_n = -\frac{J_n(ka_o)}{H_n^{(1)}(ka_o)}, \quad kb = 1.001, \quad ka_o = 0.5, \quad \phi = 30\%, \tag{5.2}$$

where $a_o$ is the radius of the particle.

With the above parameters, we found that truncating the matrix $\boldsymbol{\Psi}(s)$, with $|n| \leq 3$ and $|m| \leq 3$ in (3.15)–(3.17), led to accurate results when calculating the effective wavenumbers $S_p$, i.e. the roots of (3.30). Numerically calculating the wavenumbers $S_p$ then leads to figure 3.

The effective wavenumbers with the lowest attenuation (smallest imaginary part) contribute the most to the transmitted wave. In figure 3, we see two wavenumbers have lower attenuation then the rest, both within the dashed grey circle. The blue point represents the wavenumber that most of the literature focuses on calculating: it has a positive real part and therefore propagates forwards along the $x$-axis (into the material) as is expected for a transmitted wave. However, the other wavenumber, with negative real part, is equally as important because it actually has lower attenuation. Figure 1 illustrates several effective wavenumbers, some travelling forward into the material, while others have negative phase direction (travel backwards).

In figure 3, we see what appears to be an infinite sequence of effective wavenumbers $S_p$, where $|S_p| \to \infty$ as $p \to \infty$. To confirm their existence, and to find their locations as $|p| \to \infty$, we develop asymptotic formulae in appendix B. The results of the asymptotics are summarized below.

For *monopole scatterers*, where $n = m = 0$ in (3.15), equations (B 7) give the effective wavenumbers $S_p^o$ at leading order:

$$bS_p^{o\pm} = \sigma_p^\pm + i\log\left(\frac{|\sigma_p^\pm|^{3/2}}{r_c}\right), \quad \begin{cases} \sigma_p^+ = \theta_c + 2\pi p & \text{for } p > -\left\lceil \frac{\theta_c}{2\pi} \right\rceil, \\ \sigma_p^- = \theta_c - \frac{3\pi}{2} - 2\pi p & \text{for } p > \left\lceil \frac{\theta_c}{2\pi} - \frac{3}{4} \right\rceil \end{cases} \tag{5.3}$$

and

$$r_c e^{i\theta_c} = \sqrt{2\pi} nb^2 T_0 H_0^{(1)}(kb) e^{-\frac{i\pi}{4}}, \quad r_c > 0, \quad -\pi \leq \theta_c \leq \pi, \tag{5.4}$$

and for any integer $p$. We use the superscript 'o' to distinguish these wavenumbers for monopole scatterers from others. Even though (5.3) was deduced for large integer $p$, it gives remarkable

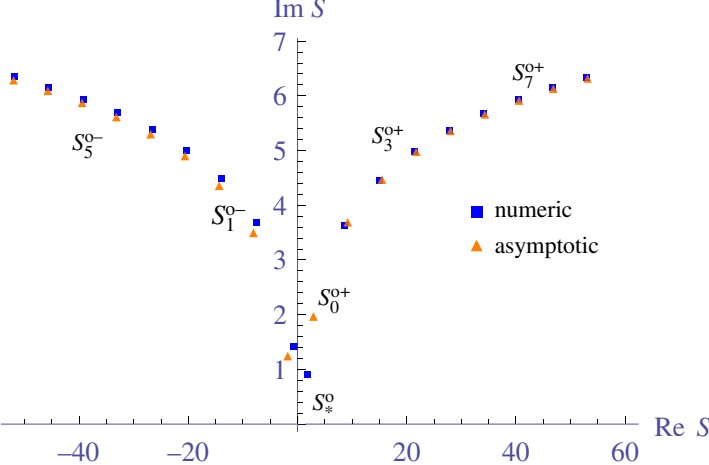

**Figure 4.** Comparison of the asymptotic formula (5.3), which predicts an infinite number of effective wavenumbers, with numerical solutions for the effective wavenumbers (5.1). The parameters used are given by (5.2), with their definitions explained in (2.4)–(2.9). Here we chose $b = 1.0$, so the non-dimensional wavenumbers $bS$ are the same as shown. The asymptotic formula is surprisingly accurate except for the two lowest attenuating wavenumbers. The wavenumber $S_*^o$ can be calculated by using low volume fraction expansions [11]. (Online version in colour.)

agreement with numerically calculated wavenumbers, except for the two lowest attenuating wavenumbers, as shown in figure 4. In the figure, we denoted $S_*^{o\pm}$ as the effective wavenumber that can be calculated by low volume fraction expansions [11,14].

For *multipole scatterers*, where both $n$ and $m$ could potentially range from $-\infty$ to $\infty$ in (3.15), we can also calculate an infinite sequence of effective wavenumbers. To show this explicitly, we consider the limit of large $bk$, with $|k| \sim |S|$. In the opposite limit $bk \ll 1$, the Rayleigh limit, only one effective wavenumber is required [44,45].

At leading order, the asymptotic solution of (B 11) leads to the effective wavenumbers:

$$bS_p^{k\pm} = \sigma_p^{\pm} + i \log\left(\frac{|\sigma_p^{\pm} - a|\sqrt{a|\sigma_p^{\pm}|}}{r_c}\right), \tag{5.5}$$

$$\begin{cases} \sigma_p^+ = \theta_c + a + 2\pi p & \text{for } p > \left\lceil -\frac{\theta_c + a}{2\pi} \right\rceil, \\ \sigma_p^- = \theta_c + a - \frac{3\pi}{2} - 2\pi p & \text{for } p > \left\lceil \frac{\theta_c + a}{2\pi} - \frac{3}{4} \right\rceil, \end{cases} \tag{5.6}$$

$$r_c e^{i\theta_c} = -2inb^2 \sum_{n=-\infty}^{\infty} T_n, \quad r_c > 0 \quad \text{and} \quad -\pi \le \theta_c \le \pi, \tag{5.7}$$

for integer $p$. This confirms that there are an infinite number of effective wavenumbers for large scatterers, i.e. $bk \gg 1$. The distribution of these wavenumbers is similar to the monopole wavenumbers shown in figure 4.

These asymptotic formulae (5.3) and (5.5) demonstrate the existence of multiple effective waves in the limit of small (monopole and Dirichlet) scatterers (5.3) and large scatterers (5.5). However, neither of these formulae, nor the low volume fraction expansions of the wavenumber [11], are able to accurately estimate the low attenuating backward travelling effective wavenumber such as $S_1$ shown in figure 3 (in this case not related to the $S_1^{o\pm}$ and $S_1^{k\pm}$ given above). There is currently no way to analytically estimate these types of wavenumbers, even though they are necessary to accurately calculate transmission due to their small attenuation. The only approach it seems is to numerically solve (3.30).

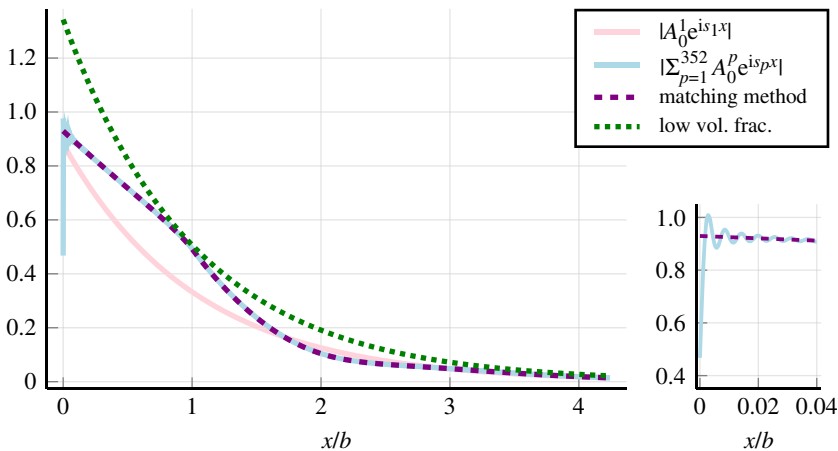

**Figure 5.** Compares the absolute value of the average field $A_0(x)$ calculated by different methods. The field $A_0(x)$ is closely related to the average transmitted wave [13]. The non-dimensional wavenumber $kb = 1.001$, the other parameters are given by (6.1), with their definitions explained by (2.4)–(2.9). Using the Wiener–Hopf solution (4.4), we approximate $A_0(x)$ by using either 352 effective wavenumbers $s_1$, $s_2$, ..., $s_{352}$, or just 1 effective wavenumber $s_1$. The Matching Method also accounts for multiple effective wavenumbers, and is described in [18]. The low volume fraction method assumes a low volume fraction expansion for just one effective wavenumber [11]. The small graph on the right is a magnification of the region around $x = 0$. Close to the boundary $x = 0$, both $A_0^1 e^{is_1 x}$ and the low volume fraction method are inaccurate, which would potentially lead to inaccurate predictions for transmission and reflection. (Online version in colour.)

## 6. Numerical results

Here we present numerical results for monopole scatterers, as these have explicit expressions for reflection (4.5) and the transmitted wave (4.4) (or more accurately the average scattering coefficients). We compare our analytic solution with a classical method that assumes only one effective wavenumber [11,13], and the Matching Method [18], recently proposed by the authors. It should be noted that all of these approaches aim to solve the same equation (2.14).

Note that for monopole scatterers, using only one effective wavenumber $s_1$ can, in some cases, lead to accurate results. However, for multipole scatterers (a more common scenario practically) this is rarely the case because, as shown by figure 3, there can be at least two effective wavenumbers with low attenuation, and therefore both are needed to obtain accurate results.

For the numerical examples, we use the parameters

$$T_0 = -\frac{J_0(ka_o)}{H_0^{(1)}(ka_o)}, \quad b = 1.001, \quad a_o = 0.5, \quad \theta_{\text{inc}} = \frac{\pi}{4}, \quad \phi = 30\%, \tag{6.1}$$

which implies that the number fraction $\mathfrak{n} \approx 0.38$ per unit area. When we choose to fix the wavenumber, as we do for figures 5 and 6, we use $bk = 1.001$. This leads to a wavelength $(2\pi/k)$ which is roughly six times larger than the particle diameter. If the particle was, say, more than 100 times smaller than the wavelength, then only one effective wavenumber in the sum (4.4) would be necessary to accurately calculate $A_0(X)$.

To start we compare the average scattering coefficient $A_0(x)$ calculated by the Wiener–Hopf solution (4.4) with other methods in figure 5. The most accurate of these other methods is the Matching Method [18,46], and it closely agrees with the Wiener–Hopf solution when using 352 effective wavenumbers. The exception is the region close to the boundary $x = 0$, where the Wiener–Hopf solution experiences a rapid transition. The low volume fraction method is the

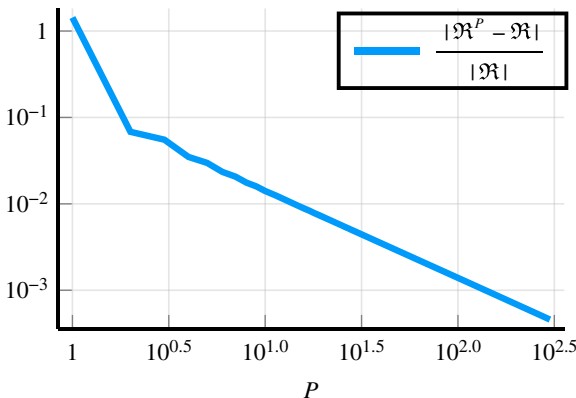

**Figure 6.** Demonstrates, with a log–log graph, how increasing the number of effective waves $P$ leads to a more accurate reflection coefficient $\mathfrak{R}^P$, when using (4.6). The non-dimensional wavenumber is $kb = 1.001$, and the other parameters used are given by (6.1), with their definitions explained by (2.4)–(2.9). Here $\mathfrak{R}$ is the reflection coefficient given by (4.5). The error $|\mathfrak{R}^P - \mathfrak{R}|$ continuously drops as $P$ increases because of the rapid transition that occurs to $A_0(x)$ near the boundary $x = 0$, see figure 5. However, methods such as the Matching Method [18] are able to accurately calculate the reflection coefficient without taking into account this rapid transition. (Online version in colour.)

most commonly used in the literature: it assumes a small particle volume fraction[10] and just one effective wavenumber [11,13]. One significant conclusion we can draw from figure 5 is that both the low volume fraction method and $A_0^1 e^{is_1 x}$ are inaccurate near the boundary $x = 0$. This means that both of these methods lead to inaccurate reflection coefficients.

In general, the Wiener–Hopf method does not lead to an explicit formula for the reflection coefficient (3.35), because we do not have an exact factorization (3.23) for any truncated square matrices. However, there are methods [13,18,20,29,30,34] to calculate $A_n(x)$, from which we can obtain the reflection coefficient (2.12). The method [18] also accounts for multiple effective wavenumbers. So one important question is: when using (2.12), how many effective wavenumbers do we need to obtain an accurate reflection coefficient?

In figure 6, we show how increasing the number of effective waves $P$ reduces the error between $\mathfrak{R}^P$ (4.6) and $\mathfrak{R}$ (4.5). To calculate a highly accurate reflection coefficient $\mathfrak{R}$, we could use either (4.5) or the Matching Method [18,46], as both give approximately the same $\mathfrak{R}$.

Now we ask: how does the reflection coefficient (4.6), deduced via the Wiener–Hopf technique, compare with other methods across a broader range of wavenumbers? The result is shown in figure 7, where $\mathfrak{R}^O$ is a low volume fraction expansion[11] of just one effective wavenumber [13]. The reflection coefficient $\mathfrak{R}^M$ is calculated from the Matching Method [18,46]. The general trend is clear: $\mathfrak{R}^O$ becomes more inaccurate as we increase the background wavenumber $kb$. On the other hand, both $\mathfrak{R}^M$ and $\mathfrak{R}$ agree closely over all $k$.

One result to note is the 'instability' exhibited by the Wiener–Hopf solution near the boundary $x = 0$, see figure 5. This instability occurs because we represented $A_0(x)$ as a superposition of truncated waves, which is only accurate as long as the discarded terms are small. So, for a truncation number $P$, we can expect the instability to occur when $e^{is_P x}$ is not small, i.e. $x \approx 1/\text{Im } s_p$. However, this instability does not affect the accuracy of the reflection coefficient (4.5) deduced by the Wiener–Hopf technique, as demonstrated by close agreement with the Matching Method in figure 7.

---

[10]For the low volume fraction method, we used a small volume fraction expansion for the wavenumber, but we numerically evaluated the wave amplitude. This is because the alternative, a small volume fraction expansion of the wave amplitude, led to poor results.

[11]We use the reflection coefficient [13, (eqn (39))], rather than the explicit low volume fraction expansion [13, (eqn (40)–(41))]. This is because using equations (40)–(41) led to roughly double the error we show.

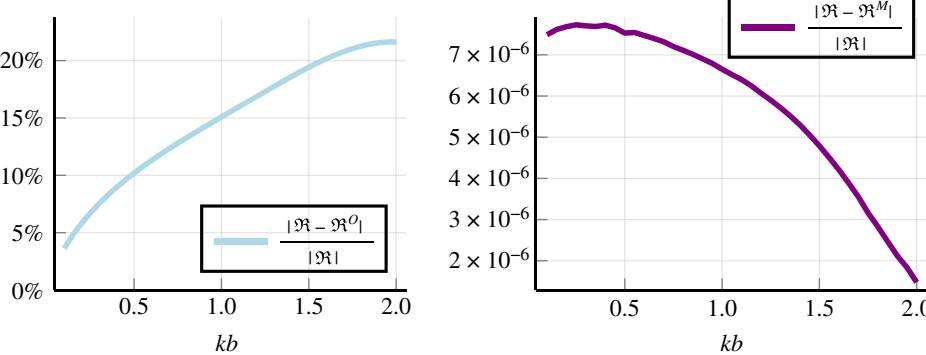

**Figure 7.** Compares different methods for calculating the reflection coefficient when varying the non-dimensional wavenumber $kb$. The other parameters used are given by (6.1), with their definitions explained in (2.4)–(2.9). Here $\mathfrak{R}$ is given by the Wiener–Hopf solution (4.5), $\mathfrak{R}^O$ uses a low volume fraction expansion of just one effective wavenumber [13], and $\mathfrak{R}^M$ is calculated from the Matching Method [18]. (Online version in colour.)

# 7. Conclusion and next steps

The major result of this paper is to prove that the ensemble-averaged field in random particulate materials consists of a superposition of waves, with complex effective wavenumbers, for one fixed incident wavenumber. These effective wavenumbers are governed by the dispersion equation (5.1) and are independent of the angle of incidence $\theta_{\text{inc}}$. We showed asymptotically in §5 that this has an infinite number of solutions, and hence there are an infinite number of effective wavenumbers. The Wiener–Hopf technique also provides a simple and elegant expression for the reflection coefficient (3.35), whose form can be used to guide and assess methods to characterize microstructure [47,48].

To numerically implement the Wiener–Hopf technique, we considered particles that scatter only in their monopole mode in §6. There we saw that when close to the interface of the half-space, a large number of effective wavenumbers were necessary to reach accurate agreement with an alternative method from the literature, the Matching Method as introduced by the authors in [18]. To obtain a constructive method via the Wiener–Hopf technique for general scatterers, and not just monopole scatterers, will require the factorization of a matrix function [31], which is challenging. For these reasons, the Matching Method [18] is presently more effective than using the Wiener–Hopf technique. However, there is ongoing work to use approximate methods [33,34,49] which exploit the symmetry and properties of the matrix (3.15).

Moving forwards, this paper together with [18], establishes accurate and robust solutions to the governing equation (2.14). These same methods can now be translated to three spatial dimensions and vectorial waves (e.g. elasticity and electromagnetics), with much of the groundwork already available [12,16,40]. Some clear challenges, that can now be addressed, are to verify the accuracy of the statistical assumptions used to deduce (2.14). These include the hole-correction and the quasi-crystalline approximations. As these are now the only assumptions used, we could compare the solution of (2.14) with multipole methods [50,51] in order to investigate their accuracy and limits of validity.

Data accessibility. We provide the code to generate all the graphs in [46].

Authors' contributions. A.L.G. and I.D.A. conceived of the study and drafted the manuscript. A.L.G., I.D.A. and W.J.P. edited the manuscript and gave final approval for publication.

Competing interests. We have no competing interests.

Funding. This work was funded by EPSRC (grant nos EP/M026205/1, EP/L018039/1 and EP/S019804/1) including via the Isaac Newton Institute (grant nos EP/K032208/1 and EP/R014604/1) and partial support from the UKAN grant EPSRC (grant no. EP/R005001/1).

# Appendix A. The Fourier transformed kernel $\hat{\psi}_n(s)$

Here we calculate the Fourier transform (3.3) of $\psi_n(X)$ (2.13). To do so, it is simpler to use

$$F_n(X, Y) = (-1)^n H_n^{(1)}(kR)e^{in\Theta}. \tag{A 1}$$

Note that both $F_n(X, Y)$ and $e^{i(sX + Yk\sin\theta_{inc})}$ satisfy wave equations, with

$$\nabla^2 F_n(X, Y) = -k^2 F_n(X, Y) \quad \text{and} \quad \nabla^2 e^{i(sX + Yk\sin\theta_{inc})} = -S^2 e^{i(sX + Yk\sin\theta_{inc})},$$

where we used (2.13)$_2$ for the first equation and (3.19) for the second equation. This means that we can use Green's second identity to obtain

$$(k^2 - S^2) \int_{\mathcal{B}} e^{i(sX + Yk\sin\theta_{inc})} F_n(X, Y) \, dXdY$$

$$= \int_{\partial\mathcal{B}} \left[ \frac{\partial e^{i(sX + YK\sin\theta_{inc})}}{\partial \boldsymbol{n}} F_n(X, Y) - e^{i(sX + KY\sin\theta_{inc})} \frac{\partial F_n(X, Y)}{\partial \boldsymbol{n}} \right] dz, \tag{A 2}$$

for any area $\mathcal{B}$ in which the integrand is analytic, where $\boldsymbol{n}$ is the outwards pointing unit normal and dz is a differential length along the boundary $\partial\mathcal{B}$. To calculate $\hat{\psi}_n(s)$, we take the region $\mathcal{B}$ to be defined by $R \geq b$, with $(R, \Theta)$ being the polar coordinates of $(X, Y)$, in which case the integral over $\mathcal{B}$ converges because as $R \to \infty$ we have that

$$|e^{i(sX + Yk\sin\theta_{inc})} F_n(X, Y)| \sim \frac{|e^{isR\cos\Theta} e^{ikR(1 + \sin\Theta\sin\theta_{inc})}|}{\sqrt{\pi |k|R/2}}$$

$$\leq \frac{|e^{-R(\text{Im}\,k(1 - |\sin\theta_{inc}|) - |\text{Im}(s)|)}|}{\sqrt{\pi |k|R/2}} \to 0, \tag{A 3}$$

exponentially fast when $|\text{Im}(s)| < \text{Im}\,k(1 - |\sin\theta_{inc}|)$. Under this restriction, and by assuming $S \neq \pm k$, (A 2) then leads to

$$\hat{\psi}_n(s) = \int_{R \geq b} e^{isX + ikY\sin\theta_{inc}} F_n(X, Y) \, dXdY = \frac{\mathcal{I}_n(b)}{k^2 - S^2}, \tag{A 4}$$

by using $sX + Yk\sin\theta_{inc} = RS\cos(\theta - \theta_S)$ from (3.19) and

$$\mathcal{I}_n(R) = \int_0^{2\pi} -\frac{\partial e^{iSR\cos(\Theta - \theta_S)}}{\partial R} F_n(kX) + e^{iSR\cos(\Theta - \theta_S)} \frac{\partial F_n(kX)}{\partial R} R \, d\Theta$$

$$= (-1)^n \int_0^{2\pi} e^{iSR\cos(\Theta - \theta_S)} e^{in\Theta} [kH_n^{(1)'}(kR) - iS\cos(\Theta - \theta_S)H_n^{(1)}(kR)] R d\theta$$

$$= (-1)^n \int_0^{2\pi} \sum_{m=-\infty}^{\infty} i^m J_m(SR) e^{im(\Theta - \theta_S)} \Big[ ke^{in\Theta} H_n^{(1)'}(kR)$$

$$- \frac{iS}{2}(e^{i(n+1)\Theta - i\theta_S} + e^{i(n-1)\Theta + i\theta_S}) H_n^{(1)}(kR) \Big] R d\Theta$$

$$= 2\pi(-i)^n Re^{in\theta_S} \left[ kJ_n(SR)H_n^{(1)'}(kR) - SJ_n'(SR)H_n^{(1)}(kR) \right], \tag{A 5}$$

where $J_n$ is the Bessel function of the first kind, and we used the Jacobi–Anger expansion on $e^{iSR\cos(\Theta - \theta_S)}$, integrated over $\Theta$ and used the identity $J_{n-1}(SR) - J_{n+1}(SR) = 2J_n'(SR)$. In summary

$$\hat{\psi}_n(s) = 2\pi \frac{(-i)^n e^{in\theta_S}}{\alpha^2 - s^2} N_n(bS), \tag{A 6}$$

when the condition (3.10) is satisfied, with $N_n$ given by (3.17).

Below we establish some useful properties for $\hat{\psi}_n(s)$. In particular, we show that $\hat{\psi}_n(s)$ has no branch-points.

The function $N_n(bS)$, for integer values of $n$, can be expanded around $S = 0$ as

$$N_n(bS) = S^{|n|} \sum_{m=0}^{\infty} c_{m|n|} S^{2m}, \tag{A 7}$$

where the $c_{m|n|}$ are some constants that depend on $m$ and $|n|$, and the radius of convergence of the series above is infinite. Using (3.19), we can write

$$e^{in\theta_S} = e^{i\mathrm{sgn}(n)|n|\theta_S} = (\cos\theta_S + \mathrm{sgn}(n)i\sin\theta_S)^{|n|} = (s + \mathrm{sgn}(n)ik\sin\theta_{\mathrm{inc}})^{|n|} S^{-|n|}. \tag{A 8}$$

Substituting (A 7) and (A 8) in (A 6) results in

$$\hat{\psi}_n(s) = \frac{2\pi(-i)^n}{\alpha^2 - s^2}(s + \mathrm{sgn}(n)ik\sin\theta_{\mathrm{inc}})^{|n|} \sum_{m=0}^{\infty} c_{m|n|} S^{2m}. \tag{A 9}$$

Because $S^2 = s^2 + k^2 \sin^2\theta_{\mathrm{inc}}$, we can immediately see from the above that $\hat{\psi}_n(s)$ has no branch-points. Additionally, we can establish the properties:

$$\hat{\psi}_n(s) = \hat{\psi}_{-n}(-s) = \hat{\psi}_n(-s)e^{2in\theta_S}(-1)^n. \tag{A 10}$$

# Appendix B. Asymptotic location of the wavenumbers

Here we explicitly calculate a sequence of effective wavenumbers $S_p$, assuming $|S_p|$ large and increasing with $p$, and $\mathrm{Im}\, S_p > 0$, that asymptotically satisfy (5.1). A key step is to approximate the terms appearing in (3.16), such as

$$J_n(bS) \sim \frac{e^{(i\pi/4)+(in\pi/2)-ibS}}{\sqrt{2\pi bS}} \quad \text{and} \quad J_n'(bS) \sim \frac{e^{-(i\pi/4)+(in\pi/2)-ibS}}{\sqrt{2\pi bS}}, \tag{B 1}$$

for large $|bS|$, where the terms $e^{ibS}$ are discarded as $\mathrm{Im}\, bS \to \infty$.

*Monopole scatterers.* The simplest case is for monopole scatterers, where $n = m = 0$ in (3.16), and the effective wavenumber $S$ satisfies

$$b^2 \det \mathbf{G} = (bS)^2 - (bk)^2 + 2\pi\mathfrak{n}b^2 T_0 N_0(bS) \sim (bS)^2 - c\sqrt{bS}e^{-ibS} = 0, \tag{B 2}$$

where $c = \sqrt{2\pi}\mathfrak{n}b^2 T_0 H_0^{(1)}(kb)e^{-(i\pi/4)}$. Here we used (B 1), and ignored terms which are algebraically smaller than $bS$. To find the root of the above, we substitute

$$bS = x + i\log y, \tag{B 3}$$

where $x$ and $y$ are real, and $|x|$ and $y$ are large with $y > 1$. This leads to

$$(x + i\log y)^{3/2} - ce^{-ix}y = 0. \tag{B 4}$$

For the logarithm and square root, we use the typical branch cut $(-\infty, 0)$ and take positive values of the functions for positive arguments. For the above to be satisfied to leading order then $x^{3/2} \sim y$, which reduces the above equation to

$$x^{3/2} \sim r_c e^{i(\theta_c - x)}y, \tag{B 5}$$

where we substituted $c = r_c e^{i\theta_c}$, for real scalars $r_c$ and $\theta_c$. Equating the real and imaginary parts of the above leads to

$$x \sim \theta_c + 2\pi p \quad \text{and} \quad y \sim \frac{1}{r_c}(\theta_c + 2\pi p)^{3/2} \quad \text{for } p > -\frac{\theta_c}{2\pi} \tag{B 6}$$

and

$$x \sim \theta_c - \frac{3\pi}{2} + 2\pi p \quad \text{and} \quad y \sim \frac{1}{r_c}(-\theta_c + \frac{3\pi}{2} - 2\pi p)^{3/2} \quad \text{for } p < \frac{3}{4} - \frac{\theta_c}{2\pi}, \tag{B 7}$$

for integers $p$. From this, we can identify that, at leading order, the effective wavenumbers are given by (5.3).

*Multipole scatterers.* With the same method used above, we can also demonstrate the existence of multiple effective wavenumbers for $n, m = -M, -M+1, \ldots, M$ in (3.16). To show this explicitly, we consider $bk$ to be the same order as $bS$, that is $|k| \sim |S|$.

By considering $bk$ large, we can approximate

$$\mathrm{H}_n^{(1)}(bk) \sim \mathrm{e}^{\mathrm{i}(bk-(\pi/4)-(n\pi/2))}\sqrt{\frac{2}{\pi bk}} \quad \text{and} \quad \mathrm{H}_n^{(1)\prime}(bk) \sim \mathrm{e}^{\mathrm{i}(bk+(\pi/4)-(n\pi/2))}\sqrt{\frac{2}{\pi bk}}, \tag{B 8}$$

combining this with (B 1) and considering $|k| \sim |S|$, the term (3.16) at leading order becomes

$$b^2 G_{mn} = d_0 \delta_{mn} + c_0 T_m, \tag{B 9}$$

where

$$d_0 = (bS)^2 - (bk)^2 \quad \text{and} \quad c_0 = 2\mathrm{n}b^2 \frac{\mathrm{i}(k+S)}{\sqrt{kS}} \mathrm{e}^{\mathrm{i}b(k-S)}.$$

By simple rearrangement of the determinant, we find that[12]

$$\det(b^2\mathbf{G}) = d_0^{2M}\left(d_0 + c_0 \sum_{m=-M}^{M} T_m\right). \tag{B 10}$$

Note that $d_0 \neq 0$, i.e. $S \neq \pm k$, was necessary to reach the condition (3.10), which was used to calculate the Fourier transforms (3.13). Taking this into consideration, and taking the limit $M \to \infty$, the effective wavenumbers $S$ must satisfy

$$d_0 + c_0 \sum_{m=-M}^{M} T_m = 0 \implies bS - bk = -2\mathrm{n}\mathrm{i}b^2 \sum_{m=-\infty}^{\infty} T_m \frac{\mathrm{e}^{\mathrm{i}b(k-S)}}{b\sqrt{kS}}. \tag{B 11}$$

Using an asymptotic expansion analogous to (B 3), the above leads to the effective wavenumbers (5.5).

## Appendix C. Equivalent determinants

For any square matrices $\mathbf{A}$ and $\mathbf{B}$, and scalar $c$, if $A_{nm} = B_{nm}c^{n-m}$ (not employing the summation convention), then

$$\det \mathbf{A} = \det \mathbf{B}. \tag{C 1}$$

This follows simply by defining the diagonal matrix $C_{nm} = \delta_{nm}c^n$, which leads to $\mathbf{A} = \mathbf{CBC}^{-1}$, and $\det(\mathbf{CBC}^{-1}) = \det \mathbf{C} \det \mathbf{B} \det \mathbf{C}^{-1} = \det \mathbf{B}$.

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
