## [Reviewer comments · Proceedings. Mathematical, Physical, and Engineering Sciences]

Review History

RSPA-2019-0344.R0 (Original submission)

Review form: Referee 1

Can the paper be shortened without overall detriment to the main message?

Yes

Do you think some of the material would be more appropriate as an electronic appendix?

No

Do you have any ethical concerns with this paper?

No

Recommendation?

Accept with minor revision (please list in comments)

Comments to the Author(s)

The manuscript demonstrates that one may have more than one effective wavenumber when describing wave propagation in an inhomogeneous medium via the effective medium (ensemble average) approach. The result is established for a half-infinite 2D space via the Wiener-Hopf

technique. Overall, the paper is interesting and well written. However, I do have a few concerns/suggestions that I hope the authors can address in the revised manuscript.

Concerns/Suggestions:

1. The assumption that the free-space (vacuum) wavenumber has a small imaginary part seems critical to all of the derived results, e.g., see eq. (3.10). Physically, k_0 (wavenumber inside a particle) may and should contain a small imaginary part but this is not the case for the free-space. I understand the mathematical necessity to add a small imaginary part to k but I wonder if one could obtain similar results for strictly real k . I suspect the answer is no and thus, the relevance of this manuscript to real physical systems is in question. It would be great if authors could elaborate on this point beyond "[The] dissipation will facilitate the use of the Wiener-Hopf technique, and after reaching the solution we can take k to be real."
2. I suspect that if k_0 has even a small imaginary component, then one will get only one dominant effective wavenumber, see fig. 3. Is this true?
3. Please state that you are using $\exp(-i \omega t)$ time factor. (This choice is consistent with the Hankel function of the 1st kind in eq. (2.10)).

Minor Typos:

1. Caption of fig. 1: θ should be replaced with θ_{inc} in the definition of k
2. p.5, 2nd line: equation 6.7 should read equation (6.7)
3. middle of p. 5: [40, Equation (15)] should read [40, equation (15)]
4. Eq. (3.1): should be $x_1 > 0$ and $x_1 \leq 0$
5. Middle of p. 12: fig. 1 illustrates several ... should read Fig. 1 illustrates several...
6. p. 14, 2nd paragraph: fig. 5 and 6 should read figs. 5 and 6

Review form: Referee 2

Can the paper be shortened without overall detriment to the main message?

Yes

Do you think some of the material would be more appropriate as an electronic appendix?

No

Do you have any ethical concerns with this paper?

No

Recommendation?

Accept with minor revision (please list in comments)

Comments to the Author(s)

The authors study the wave propagation in ensemble-averaged particulate materials and present a proof that a single wavenumber cannot describe the wave propagation. My comments are:

1. The abstract should more explicitly state the main assumptions, boundary conditions and conclusions of the paper. In particular the influence of monopole/multipole scatters (see page 15,

lines 8-12) should be stated.

2. The Matching Method should be briefly mentioned in the abstract as it forms the basis for the work presented.
3. Page 4. The spatial extent of the incident plane wave is not mentioned as an assumption that simplifies the analysis. A plane wave with spatial extent in y would encounter different boundary conditions along the boundary as the relative acoustic impedance changes, due to local changes in the particulate material. Also near and far field effects would need to be considered in the analysis.
4. Page 4, lines 47-49. The statement about the agreement between the methods is not clearly referenced. It could be read as a conclusion of the present work.
5. Page 5, line 43. The numerical order of the referencing is not correct.
6. Page 13, lines 15-16. The conclusion that the wavenumbers S_p are independent of angle of incident should be mentioned in the conclusions section.
7. Page 18, lines 15-25. Check sentence.

Decision letter (RSPA-2019-0344.R0)

26-Jul-2019

Dear Mr Gower,

On behalf of the Editor, I am pleased to inform you that your Manuscript RSPA-2019-0344 entitled "A Proof that Multiple Waves Propagate in Ensemble-Averaged Particulate Materials" has been accepted for publication subject to minor revisions in Proceedings A. Please find the referees' comments below.

The reviewer(s) have recommended publication, but also suggest some minor revisions to your manuscript. Therefore, I invite you to respond to the reviewer(s)' comments and revise your manuscript. Please note that we have a strict upper limit of 28 pages for each paper. Please endeavour to incorporate any revisions while keeping the paper within journal limits. Please note that page charges are made on all papers longer than 20 pages. If you cannot pay these charges you must reduce your paper to 20 pages before submitting your revision. Your paper has been ESTIMATED to be 21 pages. We cannot proceed with typesetting your paper without your agreement to meet page charges in full should the paper exceed 20 pages when typeset. If you have any questions, please do get in touch.

It is a condition of publication that you submit the revised version of your manuscript within 7 days. If you do not think you will be able to meet this date please let me know in advance of the due date.

To revise your manuscript, log into <https://mc.manuscriptcentral.com/prsa> and enter your Author Centre, where you will find your manuscript title listed under "Manuscripts with Decisions." Under "Actions," click on "Create a Revision." Your manuscript number has been appended to denote a revision.

You will be unable to make your revisions on the originally submitted version of the manuscript. Instead, revise your manuscript and upload a new version through your Author Centre.

When submitting your revised manuscript, you will be able to respond to the comments made by the referee(s) and upload a file "Response to Referees" in "Section 6 - File Upload". You can use

this to document any changes you make to the original manuscript. In order to expedite the processing of the revised manuscript, please be as specific as possible in your response to the referee(s).

IMPORTANT: Your original files are available to you when you upload your revised manuscript. Please delete any redundant files before completing the submission process.

In addition to addressing all of the reviewers' and editor's comments, your revised manuscript **MUST** contain the following sections before the reference list (for any heading that does not apply to your work, please include a comment to this effect):

- Acknowledgements
- Funding statement

See <https://royalsociety.org/journals/authors/author-guidelines/> for further details.

When uploading your revised files, please make sure that you include the following as we cannot proceed without these:

- 1) A text file of the manuscript (doc, txt, rtf or tex), including the references, tables (including captions) and figure captions. Please remove any tracked changes from the text before submission. PDF files are not an accepted format for the "Main Document".
- 2) A separate electronic file of each figure (tif, eps or print-quality pdf preferred). The format should be produced directly from original creation package, or original software format.
- 3) Electronic Supplementary Material (ESM): all supplementary materials accompanying an accepted article will be treated as in their final form. Note that the Royal Society will not edit or typeset supplementary material and it will be hosted as provided. Please ensure that the supplementary material includes the paper details where possible (authors, article title, journal name). Supplementary files will be published alongside the paper on the journal website and posted on the online figshare repository (<https://figshare.com>). The heading and legend provided for each supplementary file during the submission process will be used to create the figshare page, so please ensure these are accurate and informative so that your files can be found in searches. Files on figshare will be made available approximately one week before the accompanying article so that the supplementary material can be attributed a unique DOI.

Alternatively you may upload a zip folder containing all source files for your manuscript as described above with a PDF as your "Main Document". This should be the full paper as it appears when compiled from the individual files supplied in the zip folder.

Article Funder

Please ensure you fill in the Article Funder question on page 2 to ensure the correct data is collected for FundRef (<http://www.crossref.org/fundref/>).

Media summary

Please ensure you include a short non-technical summary (up to 100 words) of the key findings/importance of your paper. This will be used for to promote your work and marketing purposes (e.g. press releases). The summary should be prepared using the following guidelines:

*Write simple English: this is intended for the general public. Please explain any essential technical terms in a short and simple manner.

*Describe (a) the study (b) its key findings and (c) its implications.

*State why this work is newsworthy, be concise and do not overstate (true 'breakthroughs' are a rarity).

*Ensure that you include valid contact details for the lead author (institutional address, email address, telephone number).

Cover images

We welcome submissions of images for possible use on the cover of Proceedings A. Images should be square in dimension and please ensure that you obtain all relevant copyright permissions before submitting the image to us. If you would like to submit an image for consideration please send your image to proceedingsa@royalsociety.org

Once again, thank you for submitting your manuscript to Proceedings A and I look forward to receiving your revision. If you have any questions at all, please do not hesitate to get in touch.

Best wishes

Raminder Shergill
proceedingsa@royalsociety.org
 Proceedings A

on behalf of
 Professor Paul Smith
 Board Member
 Proceedings A

Reviewer(s)' Comments to Author:

Referee: 1

Comments to the Author(s)

The manuscript demonstrates that one may have more than one effective wavenumber when describing wave propagation in an inhomogeneous medium via the effective medium (ensemble average) approach. The result is established for a half-infinite 2D space via the Wiener-Hopf technique. Overall, the paper is interesting and well written. However, I do have a few concerns/suggestions that I hope the authors can address in the revised manuscript.

 Concerns/Suggestions:

1. The assumption that the free-space (vacuum) wavenumber has a small imaginary part seems critical to all of the derived results, e.g., see eq. (3.10). Physically, k_0 (wavenumber inside a particle) may and should contain a small imaginary part but this is not the case for the free-space. I understand the mathematical necessity to add a small imaginary part to k_0 but I wonder if one could obtain similar results for strictly real k_0 . I suspect the answer is no and thus, the relevance of this manuscript to real physical systems is in question. It would be great if authors could elaborate on this point beyond "[The] dissipation will facilitate the use of the Wiener-Hopf technique, and after reaching the solution we can take k_0 to be real."

2. I suspect that if k_0 has even a small imaginary component, then one will get only one dominant effective wavenumber, see fig. 3. Is this true?
3. Please state that you are using $\exp(-i \omega t)$ time factor. (This choice is consistent with the Hankel function of the 1st kind in eq. (2.10)).

 Minor Typos:

1. Caption of fig. 1: θ should be replaced with θ_{inc} in the definition of k
2. p.5, 2nd line: equation 6.7 should read equation (6.7)
3. middle of p. 5: [40, Equation (15)] should read [40, equation (15)]
4. Eq. (3.1): should be $x_1 > 0$ and $x_1 \leq 0$
5. Middle of p. 12: fig. 1 illustrates several ... should read Fig. 1 illustrates several...
6. p. 14, 2nd paragraph: fig. 5 and 6 should read figs. 5 and 6

Referee: 2

Comments to the Author(s)

The authors study the wave propagation in ensemble-averaged particulate materials and present a proof that a single wavenumber cannot describe the wave propagation. My comments are:

1. The abstract should more explicitly state the main assumptions, boundary conditions and conclusions of the paper. In particular the influence of monopole/multipole scatters (see page 15, lines 8-12) should be stated.
2. The Matching Method should be briefly mentioned in the abstract as it forms the basis for the work presented.
3. Page 4. The spatial extent of the incident plane wave is not mentioned as an assumption that simplifies the analysis. A plane wave with spatial extent in y would encounter different boundary conditions along the boundary as the relative acoustic impedance changes, due to local changes in the particulate material. Also near and far field effects would need to be considered in the analysis.
4. Page 4, lines 47-49. The statement about the agreement between the methods is not clearly referenced. It could be read as a conclusion of the present work.
5. Page 5, line 43. The numerical order of the referencing is not correct.
6. Page 13, lines 15-16. The conclusion that the wavenumbers S_p are independent of angle of incident should be mentioned in the conclusions section.
7. Page 18, lines 15-25. Check sentence.

Board member pre-assessment comments (if available):

This paper challenges the idea of a single effective wavenumber for an inhomogeneous material. It seems to be of very high interest and to be of wide interest to the readership of the Proceedings.

Author's Response to Decision Letter for (RSPA-2019-0344.R0)

See Appendices A & B.

Decision letter (RSPA-2019-0344.R1)

14-Aug-2019

Dear Mr Gower

I am pleased to inform you that your manuscript entitled "A Proof that Multiple Waves Propagate in Ensemble-Averaged Particulate Materials" has been accepted in its final form for publication in Proceedings A.

Our Production Office will be in contact with you in due course. You can expect to receive a proof of your article soon. Please contact the office to let us know if you are likely to be away from e-mail in the near future. If you do not notify us and comments are not received within 5 days of sending the proof, we may publish the paper as it stands.

Open access

You are invited to opt for open access, our author pays publishing model. Payment of open access fees will enable your article to be made freely available via the Royal Society website as soon as it is ready for publication. For more information about open access please visit http://royalsocietypublishing.org/site/authors/open_access.xhtml. The open access fee for this journal is ?1700/\$2380/?2040 per article. VAT will be charged where applicable.

Note that if you have opted for open access then payment will be required before the article is published? payment instructions will follow shortly.

If you wish to opt for open access then please inform the editorial office (proceedingsa@royalsociety.org) as soon as possible.

Your article has been estimated as being 21 pages long. Our Production Office will inform you of the exact length at the proof stage.

Proceedings A levies charges for articles which exceed 20 printed pages. (based upon approximately 540 words or 2 figures per page). Articles exceeding this limit will incur page charges of ?150 per page or part page, plus VAT (where applicable).

Under the terms of our licence to publish you may post the author generated postprint (ie. your accepted version not the final typeset version) of your manuscript at any time and this can be made freely available. Postprints can be deposited on a personal or institutional website, or a recognised server/repository. Please note however, that the reporting of postprints is subject to a media embargo, and that the status the manuscript should be made clear. Upon publication of the definitive version on the publisher?s site, full details and a link should be added.

You can cite the article in advance of publication using its DOI. The DOI will take the form: 10.1098/rspa.XXXX.YYYY, where XXXX and YYYY are the last 8 digits of your manuscript number (eg. if your manuscript number is RSPA-2017-1234 the DOI would be 10.1098/rspa.2017.1234).

For tips on promoting your accepted paper see our blog post: <https://blogs.royalsociety.org/publishing/promoting-your-latest-paper-and-tracking-your-results/>

On behalf of the Editor of Proceedings A, we look forward to your continued contributions to the Journal.

Sincerely,

Raminder Shergill
proceedingsa@royalsociety.org

on behalf of
Professor Paul Smith
Board Member
Proceedings A

Appendix A

PROCEEDINGS A

rspa.royalsocietypublishing.org

Research

Article submitted to journal

Subject Areas:

Wave motion, acoustics,
mathematical physics

Keywords:

wave propagation, random media,
composite materials, backscattering,
multiple scattering, ensemble
averaging, Wiener-Hopf

Author for correspondence:

Artur L. Gower

e-mail: arturgower@gmail.com

website: arturgower.github.io

A Proof that Multiple Waves Propagate in Ensemble-Averaged Particulate Materials

Artur L. Gower¹, I. David Abrahams², and William J. Parnell³

¹Department of Mechanical Engineering, The University of Sheffield, UK,

² Isaac Newton Institute for Mathematical Sciences, 20 Clarkson Road, Cambridge CB3 0EH, UK

³ School of Mathematics, University of Manchester, Oxford Road, Manchester M13 9PL, UK

Effective medium theory aims to describe a complex inhomogeneous material in terms of a few important macroscopic parameters. To characterise wave propagation through an inhomogeneous material, the most crucial parameter is the *effective wavenumber*. For this reason, there are many published studies on how to calculate a single effective wavenumber. Here we present a proof that there *does not* exist a unique effective wavenumber; instead, there are an infinite number of such (complex) wavenumbers. We show that in most parameter regimes only a small number of these effective wavenumbers make a significant contribution to the wave field. However, to accurately calculate the reflection and transmission coefficients, a large number of the (highly attenuating) effective waves is required. For clarity, we present results for scalar (acoustic) waves for a two-dimensional material filled (over a half space) with randomly distributed circular cylindrical inclusions. We calculate the effective medium by ensemble averaging over all possible inhomogeneities. The proof is based on the application of the Wiener-Hopf technique and makes no assumption on the wavelength, particle boundary conditions/size, or volume fraction. This technique provides a simple formula for the reflection coefficient, which can be explicitly evaluated for monopole scatterers. We compare results with an alternative numerical matching method.

1. Introduction

Materials comprising particles or inclusions that are randomly distributed inside a uniform host medium occur frequently in the world around us. They occur as synthetically fabricated media and also in nature. Common examples include composites, emulsions, suspensions, complex gases, and polymers. Understanding how electromagnetic, elastic, or acoustic waves propagate through these materials is necessary in order to characterise the properties of these materials, and also to design new materials that can control wave propagation.

The wave scattered from a particulate material will be influenced by the positions and properties of all particles, which are usually unknown. However, this scattered field, averaged over space or over time, depends only on the average particle properties. Many measurement systems perform averaging over space, if the receivers or incident wavelength are large enough [1], or over time [2]. In most cases, this averaging process is the same as averaging over all possible particle configurations. Such systems are sometimes called ergodic [2,3]. In this paper, we focus on ensemble averaged waves, satisfying the scalar wave equation in two-dimensions, reflecting from, and propagating in, a half-space particulate material. In certain scenarios, such as light scattering, it is easier to measure the average intensity of the wave. However, even in these cases, the ensemble-averaged field is often needed as a first step [4,5].

One driving principle, often used in the literature, is that the ensemble-averaged wave itself satisfies a wave equation with a single effective wavenumber [6–8]. Reducing an inhomogeneous material, with many unknowns, down to one effective wavenumber is attractive as it greatly reduces the complexity of the problem. For this reason many papers have attempted to deduce this unique effective wavenumber from first principles in electromagnetism [3,9,10], acoustics [11–15] and elasticity [16,17]. See [18] for a short overview of the history of this topic, including typical statistical assumptions employed within the methods, such as hole-correction and the quasi-crystalline approximation, which we also adopt here.

The assumption that the ensemble averaged wave field satisfies a wave equation, with an effective wavenumber, has never been fully justified. Here we prove that there *does not* exist a unique effective wavenumber but instead there are an infinite number of them. Gower et al. [18] first showed that there exist many effective wavenumbers, and provided a technique, the *Matching Method*, to efficiently calculate the effective wave field. In the present paper and [18], we show that for some parameter regimes, at least two effective wavenumbers are needed to obtain accurate results, when compared with numerical simulations. We also provide examples of how a single effective wave approximation leads to inaccurate results for both transmission and reflection for a halfspace filled with particles, see Figure 1.

Although the *Matching Method* developed in [18] gave accurate results, when compared to numerical methods and known asymptotic limits, the limitations of the method were not immediately clear. Here however we illustrate that the Matching Method is robust, because combining many effective wavenumbers is not just a good approximation, it is an analytical solution to the integral equation governing the ensemble averaged wave field. We prove this by employing the Wiener-Hopf technique and then, for clarity, illustrate the solution for particles that scatterer only in their monopole mode. The Wiener-Hopf technique also gives a simple and elegant expression for the reflection coefficient.

The Wiener-Hopf technique is a powerful tool to solve a diverse range of wave scattering problems, see [19, Chapter 5. Wiener-Hopf Technique] and [20,21] for an introduction. It is especially useful for semi-infinite domains [22–27] and boundary value problems of mixed type. In this work, the Wiener-Hopf technique clearly reveals the form of the analytic solution, but to compute the solution would require an analytic factorisation of a matrix-function. To explicitly perform this factorisation is difficult [28–31]. Indeed this is often the hardest aspect of employing the Wiener-Hopf technique, although there exist approximate methods for this purpose [28,32–34]. We do not focus in this article on these analytic factorisations, as there already

exists a method to compute the required solution [18]. Instead, the present work acts as proof that the Matching Method [18] faithfully reproduces the form of the analytic solution.

Figure 1. When an incident plane wave $e^{ik \cdot (x,y)}$, with $\mathbf{k} = k(\cos \theta_{inc}, \sin \theta_{inc})$, encounters an (ensemble-averaged) particulate material, it excites many transmitted plane waves and one reflected plane wave. The transmitted waves are of the form $e^{is_p \cdot (x,y)}$ with wavenumbers $s_p = S_p(\cos \theta_p, \sin \theta_p)$ where both S_p and θ_p are complex numbers. The larger $\text{Im } s_p$, the more quickly the wave attenuates as it propagates into the half-space and the smaller the drawn vector for that wave above. The results shown here represent the effective wavenumbers for parameters (5.2), which are shown in Figure 3.

Figure 1 shows the main setup and result of this paper: an incident plane wave excites the half-space $x > 0$ filled with ensemble-averaged particles (the blue region), which generates a reflected wave and many effective transmitted waves. The s_p are the transmitted wavevectors, and the smaller the length of the vector, the faster that effective wave attenuates as it propagates further into the material.

The paper begins by summarising the equations that govern ensemble averaged waves in two-dimensions in section 2. Following this, in section 3 we apply the Wiener-Hopf technique to the governing integral equation and deduce that the solution is a superposition of plane waves, each with a different effective wavenumber. A simple expression for the reflection coefficient is also derived. In section 4 we specialise the results for particles that scatter only in the monopole mode, which leads to a closed form analytic solution.

The dispersion relation (3.30), derived in section 3, admits an infinite number of solutions, the effective wavenumbers. In section 5, we deduce asymptotic forms for the effective wavenumbers in both a low and high frequency limit. In section 6 we compare numerical results for monopole scatterers, using the Wiener-Hopf technique, with classical methods that assume only one effective wavenumber [11,13], and the Matching Method introduced in [18]. In general, when comparing predicted reflection coefficients, the Wiener-Hopf and Matching Method agree well, whereas the classical single-effective-wavenumber method can disagree by anywhere up to 20%. These results are discussed in section 7 together with anticipated future steps.

2. Waves in ensemble averaged particles

Consider a region filled with particles or inclusions that are uniformly distributed. The field u is governed by the scalar wave equations:

$$\nabla^2 u + k^2 u = 0, \quad (\text{in the background material}), \quad (2.1)$$

$$\nabla^2 u + k_o^2 u = 0, \quad (\text{inside a particle}), \quad (2.2)$$

where k and k_o are the real wavenumbers of the background and inclusion materials, respectively. We assume all particles are identical, except for their position and orientation, for simplicity. For a distribution of particles, or multi-species, see [15].

Our goal is to calculate the ensemble average field $\langle u(x, y) \rangle$, that is, the field averaged over all possible particle positions and orientations. For clarity, and ease of exposition, we consider that the particles are equally likely to be located anywhere except that they cannot overlap (this is often called the *hole correction* assumption). We also assume the quasi-crystalline approximation; for details on this, and for further details on deducing the results in this section, see [11,15,18].

By splitting the total steady wave field $u(x, y)$ into a sum of the incident wave $u_{\text{inc}}(x, y)$ and waves scattered by each particle, the j th scattered wave being $u_j(x, y)$, we can write:

$$u(x, y) = u_{\text{inc}}(x, y) + \sum_j u_j(x, y). \quad (2.3)$$

A simple and useful scenario to consider is when all particles are placed only within the half-space¹ $x > 0$, which are then excited by a plane wave, with implicit time dependence $e^{i\omega t}$, incident from a homogeneous region:

$$u_{\text{inc}}(x, y) = e^{i(\alpha x + \beta y)}, \quad \text{with} \quad (\alpha, \beta) = (k \cos \theta_{\text{inc}}, k \sin \theta_{\text{inc}}), \quad (2.4)$$

where we restrict the incident angle $-\frac{\pi}{2} < \theta_{\text{inc}} < \frac{\pi}{2}$, as shown in Figure 1, and consider a slightly dissipative medium with

$$\text{Re } k > 0 \quad \text{and} \quad \text{Im } k > 0. \quad (2.5)$$

This dissipation will facilitate the use of the Wiener-Hopf technique, and after reaching the solution we can take k to be real².

To describe the particulate medium we employ the following notation:

$$b = \text{the minimum distance between particle centres}, \quad (2.6)$$

$$n = \text{number of particles per unit area}, \quad (2.7)$$

$$T_n = \text{the coefficients of the particle's T-matrix}, \quad (2.8)$$

$$\phi = \frac{\pi n b^2}{4} = \text{particle area fraction}. \quad (2.9)$$

Although the area fraction ϕ , normally called the volume fraction, is a combination of other parameters, it is useful because it is non-dimensional. If we let a_o be the maximum distance from the particle's centre to its boundary, then we can set $b = \gamma a_o$, where $\gamma \geq 2$ so as to avoid two particles overlapping. The volume fraction that does not include the exclusion zone ϕ' , as used in [18, equation (4.7)], is then $\phi' = 4\phi/\gamma^2$.

The T_n are the coefficients of a diagonal T-matrix [35–39]. The T-matrix determines how the particle scatters waves, and so depends on the particle's shape and boundary conditions. A diagonal T-matrix can be used to represent either a radially symmetric particle, or particles averaged over their orientation, assuming the orientations have a random uniform distribution.

The results of ensemble averaging (2.3) from first principals are deduced in a number of references [15,18] and so details of this procedure are omitted here for brevity. To represent the

¹The case where particles can be placed anywhere in the plane can lead to ill-defined integrals [11].

²Assuming $\text{Im } k = \epsilon > 0$, rather than $\epsilon \geq 0$, will facilitate calculating certain integrals that appear below. However, after reaching a solution, we can take the limit $\epsilon \rightarrow 0$ to recover the physically viable solution for $\text{Im } k = 0$.

ensemble averaged scattered wave from a particle, whose centre is fixed at (x_1, y_1) , we use

$$\langle u_1(x_1 + X, y_1 + Y) \rangle_{(x_1, y_1)} = \sum_{n=-\infty}^{\infty} A_n(x_1) e^{i\beta y_1} H_n^{(1)}(kR) e^{in\Theta}, \quad (2.10)$$

for $R := \sqrt{X^2 + Y^2} > b/2$, so that (X, Y) is on the outside of this particle, with (R, Θ) being the polar coordinates of (X, Y) , $H_n^{(1)}$ are Hankel functions of the first kind, and A_n is some field we want to determine³.

By choosing⁴ $x < -b$, which is outside of the region filled with particles, then taking the ensemble average on both sides of (2.3) results in equation (6.7) of [18], given by:

$$\langle u(x, y) \rangle = u_{\text{inc}}(x, y) + \Re e^{-i\alpha x + i\beta y} \quad \text{for } x < -b, \quad (2.11)$$

which is the incident wave plus an effective reflected wave with reflection coefficient:

$$\Re = e^{i\alpha x} \sum_{n=-\infty}^{\infty} \int_0^{\infty} A_n(x_1) \psi_n(x_1 - x) dx_1, \quad (2.12)$$

where we assumed particles are distributed according to a uniform distribution, and the kernel ψ_n is given by

$$\psi_n(X) = \int_{Y^2 > b^2 - X^2}^{\infty} e^{i\beta Y} (-1)^n H_n^{(1)}(kR) e^{in\Theta} dy. \quad (2.13)$$

Later we show that, as expected, \Re is independent of x .

The system governing $A_m(x)$ is given by equation (4.7) of [18]:

$$\begin{aligned} nT_m \sum_{n=-\infty}^{\infty} \int_0^{\infty} A_n(x_2) \psi_{n-m}(x_2 - x_1) dx_2 \\ = A_m(x_1) - e^{i\alpha x_1} T_m e^{im(\pi/2 - \theta_{\text{inc}})}, \quad \text{for } x_1 \geq 0, \end{aligned} \quad (2.14)$$

for all integers m . Kristensson [40, equation (15)] presents an equivalent integral equation for electromagnetism and particles in a slab.

Our main aim is to reach an exact solution for $A_n(x)$ by employing the Wiener-Hopf technique to (2.14). We show how this also leads to simple solutions for the reflection coefficient by using (2.11). We acknowledge the authors of [11], as they noticed that (2.14) is a Wiener-Hopf integral equation; but apparently did not follow the steps indicated in the following sections⁵.

3. Applying the Wiener-Hopf technique

Equation (2.14) is convolution integral equation with a difference kernel. This means applying a Fourier transforms can lead to elegant and simple solutions. To facilitate, we must analytically extend (2.14) for all $x_1 \in \mathbb{R}$ by defining

$$\begin{aligned} nT_m \sum_{n=-\infty}^{\infty} \int_0^{\infty} A_n(x_2) \psi_{n-m}(x_2 - x_1) dx_2 \\ = \begin{cases} A_m(x_1) - e^{i\alpha x_1} T_m e^{im(\pi/2 - \theta_{\text{inc}})}, & x_1 \geq 0, \\ D_m(x_1), & x_1 < 0, \end{cases} \end{aligned} \quad (3.1)$$

for integers m , where if the $A_n(x)$ were known for $x > 0$, then the $D_n(x)$ would be given from the left hand-side. Note that the kernel ψ_n defined in (2.13) is already analytic in the domain \mathbb{R} .

³The factor $e^{i\beta y_1}$ appears due to the translational invariance of $\langle u_1(x_1, y_1) \rangle$ in y_1 , which is a result of the material being statistically homogeneous, see [15] for details.

⁴We define the reflection coefficient only for $x < -b$, instead of $x < -b/2$, so that we can use ψ_n in the formula for \Re , which will in turn facilitate calculating \Re .

⁵However, they were unable to solve it because, it seems, of a mistake in the integrand of equation (37) of [11]; they used $e^{i\beta Y}$ where they should have used $\cos(\beta Y)$.

The field $D_0(x)$ is not just an abstract construct, it is closely related to the reflected wave: by directly comparing (3.1) with the reflection coefficient (2.12), for $x < -b$, we find that

$$D_0(x) = T_0 \mathfrak{R} e^{-i\alpha x}. \quad (3.2)$$

To solve (3.1) we employ the Fourier transform and its inverse, which we define as

$$\hat{f}(s) = \int_{-\infty}^{\infty} f(x) e^{isx} dx \quad \text{with} \quad f(x) = \frac{1}{2\pi} \int_{-\infty}^{\infty} \hat{f}(s) e^{-isx} ds, \quad (3.3)$$

for any smooth function f . We then define

$$\hat{A}_n^+(s) = \int_0^{\infty} A_n(x) e^{isx} dx, \quad \hat{D}_n^-(s) = \int_{-\infty}^0 D_n(x) e^{isx} dx. \quad (3.4)$$

We can determine where \hat{A}_n^+ and \hat{D}_n^- are analytic by assuming⁶ that

$$|A_n(x)| < e^{-xc} \quad \text{for } x \rightarrow \infty, \quad (3.5)$$

$$|D_n(x)| < e^{xc} \quad \text{for } x \rightarrow -\infty, \quad (3.6)$$

for some (possibly small) positive constant c . This leads to $\hat{A}_n^+(s)$ being analytic for $\text{Im } s > -c$, while $\hat{D}_n^-(s)$ is analytic for $\text{Im } s < c$. In other words, both $\hat{A}_n^+(s)$ and $\hat{D}_n^-(s)$ are analytic in the overlapping strip

$$|\text{Im } s| < c. \quad (3.7)$$

To apply the Wiener-Hopf technique we also need to specify the large s behaviour for both $\hat{A}_n^+(s)$ and $\hat{D}_n^-(s)$. To achieve this, we assume, on physical grounds, that $A_n(x)$ is bounded when $x \rightarrow 0^+$, and $D_n(x)$ is bounded when $x \rightarrow 0^-$. Then, it can be shown [21,41] that

$$\hat{A}_n^+(s) = \mathcal{O}(|s|^{-1}) \quad \text{and} \quad \hat{D}_n^-(s) = \mathcal{O}(|s|^{-1}) \quad \text{for } |s| \rightarrow \infty, \quad (3.8)$$

in their respective half-planes of analyticity.

Applying a Fourier transform to both sides of equation (3.1), the left-hand side becomes

$$\int_0^{\infty} A_n(x_2) \int_{-\infty}^{\infty} \psi_{n-m}(x_1 - x_2) e^{isx_1} dx_1 dx_2 = \hat{A}_n^+(s) \hat{\psi}_{n-m}(s), \quad (3.9)$$

in which $\hat{\psi}_n(s)$ is well defined (i.e. analytic) for s in the strip:

$$|\text{Im } s| < (1 - |\sin \theta_{\text{inc}}|) \text{Im } k, \quad (3.10)$$

see appendix A for details. The right-hand side of (3.1) becomes

$$\int_{-\infty}^0 D_m(x_1) e^{isx_1} dx_1 + \int_0^{\infty} A_m(x_1) e^{isx_1} dx_1 - e^{im(\pi/2 - \theta_{\text{inc}})} T_m \int_0^{\infty} e^{ix_1(s+\alpha)} dx_1 = \hat{D}_m^-(s) + \hat{A}_m^+(s) - T_m \frac{ie^{im(\pi/2 - \theta_{\text{inc}})}}{(s+\alpha)^+}, \quad (3.11)$$

where for the last step we assumed $\text{Im}(s+\alpha) > 0$, which is why we use the superscript $+$ on $(s+\alpha)^+$. This assumption, together with (3.7) and (3.10), is satisfied if

$$|\text{Im } s| < \epsilon, \quad \text{where} \quad \epsilon = \min\{c, (1 - |\sin \theta_{\text{inc}}|) \text{Im } k, \text{Im } \alpha\}. \quad (3.12)$$

If (3.12) is satisfied then we can combine (3.9), (3.11) and (A 6), to obtain the Fourier transform of (3.1) in matrix form:

$$\frac{\Psi(s) \hat{A}^+(s)}{s^2 - \alpha^2} = -\hat{D}^-(s) + \frac{B}{s + \alpha}, \quad (3.13)$$

⁶The solutions for $A_n(x)$ and $D_n(x)$, in the next section, show that these assumptions do hold.

where $\hat{\mathbf{A}}^+(s)$ and $\hat{\mathbf{D}}^-(s)$ are vectors with components $\hat{A}_n^+(s)$ and $\hat{D}_n^-(s)$, respectively and

$$B_m = iT_m e^{im(\pi/2 - \theta_{\text{inc}})}, \quad (3.14)$$

$$\Psi_{mn}(s) = G_{mn}(S)(-i)^{n-m} e^{i(n-m)\theta_S}, \quad (3.15)$$

$$G_{mn}(S) = (s^2 - \alpha^2)\delta_{mn} + 2\pi n T_m N_{n-m}(bS), \quad (3.16)$$

$$N_m(bS) = bk J_m(bS) H_m^{(1)'}(bk) - bS J_m'(bS) H_m^{(1)}(bk). \quad (3.17)$$

where, for reference,

$$\Psi_{mn}(s) = (s^2 - \alpha^2) \left[\delta_{mn} - n T_m \hat{\psi}_{n-m}(s) \right], \quad (3.18)$$

and $\hat{\psi}_{n-m}(s)$ is given by (A 6). In the above θ_S and S are chosen to satisfy

$$s = S \cos \theta_S \quad \text{with} \quad S \sin \theta_S = k \sin \theta_{\text{inc}}. \quad (3.19)$$

Later we identify S and θ_S as the effective wavenumber and transmission angle. The above does not determine the sign of S for any given complex s . To fully determine S and θ_S , we take $\text{sgn}(\text{Re } s) = \text{sgn}(\text{Re } S)$ which together with (3.19) leads to

$$\theta_S = \arctan \left(\frac{k \sin \theta_{\text{inc}}}{s} \right), \quad S = \sqrt{s^2 + (k \sin \theta_{\text{inc}})^2}, \quad (3.20)$$

where both S and θ_S , when considered as functions⁷ of s , contain branch-points at $s = \pm ik \sin \theta_{\text{inc}}$ with finite branch-cut running between $-ik \sin \theta_{\text{inc}}$ and $ik \sin \theta_{\text{inc}}$. However, $\Psi(s)$ is an entire matrix function having only zeros in s and no branch-points; see the end of appendix A for details.

Determining the roots of $\det \Psi(s) = 0$ will be a key step in solving (3.13), and so the following identities will be useful

$$\Psi_{mn}(-s) T_n = \Psi_{mn}(s) T_n (-1)^{m-n} e^{2i(m-n)\theta_S} = \Psi_{nm}(s) T_m, \quad (3.21)$$

$$\det \Psi(-s) = \det \Psi(s) \quad \text{and} \quad \det \Psi(s) = \det \mathbf{G}(S). \quad (3.22)$$

where (3.21) results from (3.18) and (A 10). Equation (3.22) then follows from using (3.21)₁, (3.15), and appendix C.

(a) Multiple waves solution

To solve (3.13), we use a matrix product factorisation [42] of the form:

$$\Psi(s) = \Psi^-(s) \Psi^+(s), \quad (3.23)$$

where $\Psi^-(s)$, and its inverse, are analytic in $\text{Im } s < \epsilon$, and $\Psi^+(s)$, and its inverse, are analytic for $\text{Im } s > -\epsilon$. See (3.12) for the definition of ϵ .

For our purposes, it is enough to know that such a factorisation exists [42], as this will lead to a proof that $\mathbf{A}(x)$ is a sum of attenuating plane waves.

Multiplying both sides of (3.13) by $[\Psi^-(s)]^{-1}$ and by $(s - \alpha)_-$ leads to

$$\frac{\Psi^+(s) \hat{\mathbf{A}}^+(s)}{(s + \alpha)_+} = -(s - \alpha)_- [\Psi^-(s)]^{-1} \hat{\mathbf{D}}^-(s) + [\Psi^-(s)]^{-1} \mathbf{B} \frac{(s - \alpha)_-}{(s + \alpha)_+}, \quad (3.24)$$

where $(s + \alpha)_+$ is analytic for $\text{Im } s > -\text{Im } \alpha$, while $(s - \alpha)_-$ is analytic for $\text{Im } s < \text{Im } \alpha$. We need to rewrite the last term above as a sum of a function which is analytic in the upper half-plane (Im

⁷With our choice of branch-cut, the simplest way to compute S , with most software packages, is to use $S = \text{sgn}(\text{Re } s) \sqrt{s^2 + (k \sin \theta_{\text{inc}})^2}$ with the default cut location for the square root function, i.e. along the real negative line.

$s > -\epsilon$) and another analytic in the lower half-plane. This is achieved below:

$$[\Psi^-(s)]^{-1} \mathbf{B} \frac{(s-\alpha)_-}{(s+\alpha)_+} = - \underbrace{\frac{2\alpha}{(s+\alpha)_+} [\Psi^-(-\alpha)]^{-1} \mathbf{B}}_{\mathbf{g}^+(s)} + \underbrace{[\Psi^-(s)]^{-1} \mathbf{B} \frac{(s-\alpha)_-}{(s+\alpha)_+} + [\Psi^-(-\alpha)]^{-1} \mathbf{B} \frac{2\alpha}{(s+\alpha)_+}}_{\mathbf{g}^-(s)}, \quad (3.25)$$

where we define

$$\lim_{s \rightarrow -\alpha} \mathbf{g}^-(s) = \left[\mathbf{I} + 2\alpha [\Psi^-(-\alpha)]^{-1} \frac{d\Psi^-}{ds}(-\alpha) \right] [\Psi^-(-\alpha)]^{-1} \mathbf{B},$$

so that $\mathbf{g}^-(s)$ does not have a pole at $s = -\alpha$ and is therefore analytic for $\text{Im } s < \epsilon$.

Substituting (3.25) into (3.24) leads to

$$\frac{\Psi^+(s) \hat{\mathbf{A}}^+(s)}{(s+\alpha)_+} + \mathbf{g}^+(s) = -(s-\alpha)_- [\Psi^-(s)]^{-1} \hat{\mathbf{D}}^-(s) + \mathbf{g}^-(s). \quad (3.26)$$

Because both sides are analytic in the strip $|\text{Im } s| < \epsilon$, we can equate each side to $\mathbf{E}(s)$, some analytic function in the strip. Further, as the left-hand side (right-hand side) of (3.26) is analytic for $\text{Im } s > \epsilon$ ($\text{Im } s < -\epsilon$), we can analytically continue $\mathbf{E}(s)$ for all s , i.e. $\mathbf{E}(s)$ is entire.

To determine $\mathbf{E}(s)$ we need to estimate its behaviour as $|s| \rightarrow \infty$. From (3.8) we have that $\mathbf{A}^+(s) = (|s|^{-1})$ as $|s| \rightarrow \infty$ in the upper half-plane, and from (3.15 - 3.17):

$$\Psi(s) = (s^2 - \alpha^2) \mathbf{I} + \mathcal{O}(|s|) \quad \text{as } |s| \rightarrow \infty, \quad (3.27)$$

for s in the strip (3.12). From this we know that the factors $\Psi^+(s)$ and $\Psi^-(s)$ must be $\mathcal{O}(|s|)$ as $|s| \rightarrow \infty$, in their respective half-planes of analyticity [28]. So, the left hand-side of (3.26) behaves as $\mathcal{O}(|s|^{-1})$ as $|s| \rightarrow \infty$ in $\text{Im } s > -\epsilon$. We can therefore use Liouville's theorem to conclude that $\mathbf{E}(s) \equiv 0$, which means the Wiener-Hopf equation (3.26) is formally equivalent to

$$\hat{\mathbf{A}}^+(s) = -2\alpha [\Psi^+(s)]^{-1} [\Psi^-(-\alpha)]^{-1} \mathbf{B}, \quad (3.28)$$

$$\hat{\mathbf{D}}^-(s) = \frac{\Psi^-(s) \mathbf{g}^-(s)}{(s-\alpha)_-}. \quad (3.29)$$

Let $\mathbf{C}^+(s)$ be the cofactor matrix of $\Psi^+(s)$, so that

$$[\Psi^+(s)]^{-1} = \frac{[\mathbf{C}^+(s)]^T}{\det(\Psi^+(s))}.$$

From the property (3.22)₁ we can write $\det \Psi(s) = f(s^2)$ for some function f . Then, for every root $s = s_p$ of $\det \Psi(s)$, with $\text{Im } s_p > 0$, we have that $-s_p$ is also a root, and vice-versa. From here onwards we assume:

$$\det \Psi(s_p) = \det \Psi(-s_p) = 0 \quad \text{with } \text{Im } s_p > 0 \quad \text{and } p = 1, 2, \dots, \infty. \quad (3.30)$$

For any truncated matrix $\Psi(s)$, i.e. evaluating $m, n = -M, \dots, M$ in (3.15), the roots s_p are discrete. In section 5 we demonstrate asymptotically that they are indeed discrete for the limits of low and high wavenumber k . For the numerical results presented in this paper, we numerically solve the above dispersion relation for the truncating the matrix $\Psi(s)$, and then increase M until the roots converge (typically no more than $M = 4$ was required).

Given $\det \Psi(s) = \det \Psi^-(s) \det \Psi^+(s)$, every root of $\det \Psi(s)$ must either be a root of $\det \Psi^-(s)$ or a root of $\det \Psi^+(s)$. For $[\Psi^+(s)]^{-1}$ to be analytic in the upper half-plane, $\det \Psi^+(s)$ must only have roots $s = -s_p$. As a consequence, $\det \Psi^-(s)$ only has roots $s = s_p$.

To use the residue theorem below, we need to calculate $\det \Psi^+(s)$ for s close to the root $-s_p$, in the form

$$\begin{aligned} \det \Psi^+(s) &= \det \Psi^+(-s_p) + (s + s_p) \frac{d \det \Psi^+}{ds}(-s_p) + \mathcal{O}((s + s_p)^2) \\ &= \frac{s + s_p}{\det \Psi^-(-s_p)} \frac{d \det \Psi}{ds}(-s_p) + \mathcal{O}((s + s_p)^2) \end{aligned} \quad (3.31)$$

where we use $\frac{d \det \Psi}{ds}(-s_p)$ instead of $\frac{d \det \Psi^+}{ds}(-s_p) \det \Psi^-(-s_p)$, because it is more difficult to numerically evaluate $\frac{d \det \Psi^+}{ds}(-s_p)$.

Figure 2. An illustration of the contour integral over \mathcal{C}_D , used to calculate (3.34) for $x < 0$, and the contour integral over \mathcal{C}_A , used to calculate (3.32) for $x > 0$. The $-s_p$ (the red points) are roots of (3.30), and also the poles of (3.28). The single blue point α is the only pole of (3.29).

Using the above, and that $\mathbf{C}^+(s)$ is analytic for $\text{Im } s > -\epsilon$, we can apply an inverse Fourier transform (3.3)₂ to both sides of (3.28) and using residue calculus we find

$$\mathbf{A}(x) = -\frac{\alpha}{\pi} \int_{-\infty}^{\infty} \frac{[\mathbf{C}^+(s)]^T [\Psi^-(-\alpha)]^{-1} \mathbf{B}}{\det \Psi^+(s)} e^{-isx} ds = \begin{cases} \sum_{p=1}^{\infty} \mathbf{A}^p e^{is_p x}, & x > 0, \\ 0, & x < 0, \end{cases} \quad (3.32)$$

$$\text{with } \mathbf{A}^p = 2\alpha i \frac{\det \Psi^-(-s_p)}{\frac{d \det \Psi}{ds}(-s_p)} [\mathbf{C}^+(-s_p)]^T [\Psi^-(-\alpha)]^{-1} \mathbf{B}. \quad (3.33)$$

For $x > 0$, the integral over $s \in [-\infty, \infty]$ in (3.32) is, by Jordan's lemma, the same as a clockwise integral over the closed contour \mathcal{C}_A which surrounds the poles $-s_1, -s_2, \dots$, i.e. roots of (3.30), as shown by Figure 2. Note that the cofactor matrix $\mathbf{C}^+(s)$ contains no poles and so does not contribute additional residual terms. The yellow striped region in Figure 2 is the domain where Ψ is analytic. On the other hand, for $x < 0$, the integral (3.32) is the same as an integral over the counter-clockwise closed contour within the region $\text{Im } s > 0$ (not shown in Figure 2). The integrand has no poles in this domain and hence evaluates to zero.

Likewise, by applying an inverse Fourier transform to (3.29), we obtain:

$$\mathbf{D}(x) = \frac{1}{2\pi} \int_{-\infty}^{\infty} \frac{\Psi^-(s) \mathbf{g}^-(s)}{(s - \alpha)_-} e^{-isx} ds = \begin{cases} i \Psi^-(\alpha) [\Psi^-(-\alpha)]^{-1} \mathbf{B} e^{-i\alpha x}, & x < 0, \\ 0, & x > 0, \end{cases} \quad (3.34)$$

For $x < 0$ the above integral is the same as a counter-clockwise closed integral over \mathcal{C}_D which surrounds the pole $s = \alpha$ (recalling that $\text{Im } \alpha > 0$), as shown in Figure 2. The result is just the

residue at this pole. That is, the function $\Psi^-(s)g^-(s)$ contains no other singularities within $\text{Im } s > 0$. On the other hand, for $x > 0$ the integral is the same as a closed clockwise integral around the region $\text{Im } s < 0$ which evaluates to zero, as there are no singularities in this region (not shown in Figure 2).

Clearly (3.32) shows that $A(x)$ is a sum of plane waves with different effective wavenumbers s_p , each satisfying (3.30). In section 5 we discuss these roots in more detail, and in section 6, we see that usually only a few effective wavenumbers are required to obtain accurate results.

(b) Reflection coefficient

By substituting (3.34) in (3.2) leads to

$$\mathfrak{R} = iT_0^{-1} \sum_{n,m=-\infty}^{\infty} \Psi_{0n}^-(\alpha) [\Psi^-(-\alpha)]_{nm}^{-1} B_m. \quad (3.35)$$

Alternatively, the reflection coefficient can be calculated from (2.12) by employing the form of $A(x)$ from (3.32), which is the more common approach. To simplify, we use

$$\psi_n(X) = (-1)^n \int_{-\infty}^{\infty} e^{ikY \sin \theta_{\text{inc}}} H_n^{(1)}(kR) e^{in\Theta} dY = \frac{2}{\alpha} i^n e^{-in\theta_{\text{inc}}} e^{i\alpha X} \quad \text{for } X > 0, \quad (3.36)$$

which then implies that $\psi_n(x_1 - x) = \frac{2}{\alpha} i^n e^{-in\theta_{\text{inc}}} e^{i\alpha(x_1 - x)}$ for $x_1 \geq x$. The above is shown in [43, equation (37)] and [11, equation (65)]. This result together with (3.32) substituted into (2.12) leads to the form

$$\mathfrak{R} = \frac{2n}{\alpha} \sum_{n=-\infty}^{\infty} i^n e^{-in\theta_{\text{inc}}} \int_0^{\infty} A_n(x_1) e^{i\alpha x_1} dx_1 = \frac{2in}{\alpha} \sum_{n=-\infty}^{\infty} \sum_{p=1}^{\infty} i^n e^{-in\theta_{\text{inc}}} \frac{A_n^p}{s_p + \alpha}, \quad (3.37)$$

where we used that $\text{Im } s_p > 0$. The above agrees with [13, equation (39)] and⁸ [18, equation (6.9)].

4. Monopole scatterers

For particles that scatter only in their monopole mode, i.e. the scattered waves are angularly symmetric about each particle, we can easily calculate the factorisation (3.23). This type of scattered wave tends to dominate in the long wavelength limit for scatterers with Dirichlet boundary conditions. In acoustics, these correspond to particles with low density or low sound speed.

Once we know the factorisation (3.23), we can then calculate the average scattering coefficient (3.32) and average reflection coefficient (3.35). We will compare both of these against predictions from other methods in section 6.

(a) Wiener-Hopf factorisation

For scalar problems, there are well known techniques to factorise $\Psi_{00}(s) = \Psi_{00}^-(s)\Psi_{00}^+(s)$, such as Cauchy's integral formulation, for details see [19, Section 5. Wiener-Hopf Technique] and [21].

For monopole scatterers we use $S^2 - k^2 = s^2 - \alpha^2$ and rewrite

$$\Psi_{00}(s) = (s^2 - \alpha^2)q(s), \quad \text{with } q(s) = 1 + 2\pi n \frac{T_0 N_0 (bS)}{S^2 - k^2},$$

⁸When taking a zero thickness boundary layer, i.e. $J = 0$, and appropriate substitutions.

with $N_0(bS)$ given by (3.17). Then, because $q(s) \rightarrow 1$ as $|s| \rightarrow \infty$, we can factorise $q(s) = q^-(s)q^+(s)$ using

$$q^+(s) = \exp\left(\frac{1}{2\pi i} \int_{-\infty}^{\infty} \frac{\log q(z)}{z-s} dz\right), \quad (4.1)$$

$$q^-(s) = \exp\left(-\frac{1}{2\pi i} \int_{-\infty}^{\infty} \frac{\log q(z)}{z-s} dz\right), \quad (4.2)$$

where the integral path for $q^+(s)$ ($q^-(s)$) has to be in the strip where $q(s)$ is analytic, with the path for $q^+(s)$ ($q^-(s)$) passing below (above) z . We then have⁹ that

$$\Psi_{00}^+(s) = (s+\alpha)_+ q^+(s), \quad \Psi_{00}^-(s) = (s-\alpha)_- q^-(s), \quad \Psi_{00}^-(-s) = -\Psi_{00}^+(s), \quad (4.3)$$

where (4.3)₃ holds if $-s$ is below the integration path of (4.2) and s is above the integration path of (4.1). From (3.32) we see that we need only evaluate $\Psi_{00}^+(s)$, and therefore $q^+(s)$, for $s = s_1, s_2, \dots, s_p$ where as p increases, the s_p become more distant from the real line. Then for large z , by inspection of (3.17), we have that

$$\left| \frac{\log q(z)}{z-s} \right| \sim \frac{1}{z^{3/2} |z-s|},$$

and therefore we can accurately approximate the integral (4.1) by truncating the integration domain for large z .

(b) Explicit solution for monopole scatterers.

For monopole scatterers $A_n(x) = D_n(x) = 0$ for $|n| > 0$. Using this in (3.14 - 3.17) leads to all vectors and matrices having only one component, given by setting $n = m = 0$. In this case \mathbf{A} (3.32) reduces to

$$A_0(x) = \sum_{p=1}^{\infty} A_0^p e^{is_p x} \quad \text{with} \quad A_0^p = \frac{2\alpha T_0}{\Psi_{00}^+(\alpha)} \frac{\Psi_{00}^+(s_p)}{\frac{d\Psi_{00}}{ds}(s_p)} = \frac{T_0}{s_p - \alpha} \frac{q^+(s_p)}{q'(s_p)}, \quad (4.4)$$

for $x > 0$, where we used (4.3), $\mathbf{C}^+(s) = 1$, $\mathbf{B} = iT_0$, and $\frac{d\Psi_{00}}{ds}(-s) = -\frac{d\Psi_{00}}{ds}(s)$ for every s . Likewise for (3.35) we arrive at

$$\mathfrak{R} = \frac{\Psi_{00}(\alpha)}{(\Psi_{00}^+(\alpha))^2} = \frac{\pi n T_0 N_0(b\alpha)}{2(\alpha q^+(s))^2}. \quad (4.5)$$

Alternatively, using (3.37), we can calculate the contribution of P effective waves to the reflection coefficient

$$\mathfrak{R}^P = \frac{2i n}{\alpha} \sum_{p=1}^P \frac{A_0^p}{s_p + \alpha} = \frac{2i n T_0}{\alpha q^+(\alpha)} \sum_{p=1}^P \frac{1}{s_p^2 - \alpha^2} \frac{q^+(s_p)}{q'(s_p)} \quad \text{with} \quad \mathfrak{R} = \lim_{P \rightarrow \infty} \mathfrak{R}^P, \quad (4.6)$$

where the error $|\mathfrak{R}^P - \mathfrak{R}|$ then indicates how many effective waves are needed to accurately describe the field near the boundary $x = 0$.

5. Multiple effective wavenumbers

Equation (3.32) clearly shows that $\mathbf{A}(x)$ is a sum of attenuating plane waves, each with a different effective wavenumber s_p . These s_p satisfy the dispersion equation (3.30):

$$\det \Psi(s_p) = \det \mathbf{G}(S_p) = 0, \quad (5.1)$$

with Ψ given by (3.16) and the first identity follows from (3.22).

⁹Note that the factors $q^+(s)$ and $q^-(s)$ are singularity and pole free in their respective regions of analyticity, and so their inverses $[q^+(s)]^{-1}$ and $[q^-(s)]^{-1}$ have the same property.

Figure 3. Examples for effective wavenumbers S_p which satisfy the dispersion equation (5.1) with the properties (5.2). The blue points represent waves travelling forwards (i.e. deeper into the material), while the red represent waves travelling backwards. All these waves are excited in a reflection experiment. Two wavenumbers in particular stand out as having the lowest attenuation S_1 and S_2 , both inside the grey dashed circle. The graph on the right is a magnification of the region close to these two wavenumbers. Out of these two, most efforts in the literature have focused on calculating S_2 , as it often has the lowest imaginary part; however for this case, because S_1 has a smaller attenuation it will have a significant contribution to both transmission and reflection.

An important conclusion from $\det \mathbf{G}(S_p) = 0$ is that the wavenumbers S_p are independent of the angle of incidence θ_{inc} . We focus on showing the results for S_p , rather than s_p , because then we do not need to specify θ_{inc} .

As a specific example, let us consider circular particles with Dirichlet boundary conditions (i.e. particles with zero density or soundspeed), and the parameters

$$T_n = -\frac{J_n(ka_o)}{H_n^{(1)}(ka_o)}, \quad kb = 1.001, \quad ka_o = 0.5, \quad \phi = 30\%, \quad (5.2)$$

where a_o is the radius of the particle.

With the above parameters, we found that truncating the matrix $\Psi(s)$, with $|n| \leq 3$ and $|m| \leq 3$ in (3.15-3.17), led to accurate results when calculating the effective wavenumbers S_p , i.e. the roots of (3.30). Numerically calculating the wavenumbers S_p then leads to Figure 3.

The effective wavenumbers with the lowest attenuation (smallest imaginary part) contribute the most to the transmitted wave. In Figure 3 we see two wavenumbers have lower attenuation than the rest, both within the dashed grey circle. The blue point represents the wavenumber that most of the literature focuses on calculating: it has a positive real part and therefore propagates forwards along the x -axis (into the material) as is expected for a transmitted wave. However, the other wavenumber, with negative real part, is equally as important because it actually has lower attenuation. Figure 1 illustrates several effective wavenumbers, some travelling forward into the material, while others have negative phase direction (travel backwards).

In Figure 3 we see what appears to be an infinite sequence of effective wavenumbers S_p , where $|S_p| \rightarrow \infty$ as $p \rightarrow \infty$. To confirm their existence, and to find their locations as $|p| \rightarrow \infty$, we develop asymptotic formulas in appendix B. The results of the asymptotics are summarised below.

For *monopole scatterers*, where $n = m = 0$ in (3.15), equations (A 7) give the effective wavenumbers S_p^o at leading order:

$$bS_p^{o\pm} = \sigma_p^\pm + i \log \left(\frac{|\sigma_p^\pm|^{3/2}}{r_c} \right), \quad \begin{cases} \sigma_p^+ = \theta_c + 2\pi p & \text{for } p > -\left\lceil \frac{\theta_c}{2\pi} \right\rceil, \\ \sigma_p^- = \theta_c - \frac{3\pi}{2} - 2\pi p & \text{for } p > \left\lceil \frac{\theta_c}{2\pi} - \frac{3}{4} \right\rceil, \end{cases} \quad (5.3)$$

$$r_c e^{i\theta_c} = \sqrt{2\pi n} b^2 T_0 H_0^{(1)}(kb) e^{-\frac{i\pi}{4}}, \quad r_c > 0, \quad -\pi \leq \theta_c \leq \pi, \quad (5.4)$$

and for any integer p . We use the superscript “o” to distinguish these wavenumbers for monopole scatterers from others. Even though (5.3) was deduced for large integer p , it gives remarkably agreement with numerically calculated wavenumbers, except for the two lowest attenuating wavenumbers, as shown in Figure 4. In the figure we denoted $S_*^{\circ\pm}$ as the effective wavenumber that can be calculated by low volume fraction expansions [11,14].

Figure 4. Comparison of the asymptotic formula (5.3), which predicts an infinite number of effective wavenumbers, with numerical solutions for the effective wavenumbers (5.1). The parameters used are given by (5.2), with their definitions explained in (2.4–2.9). Here we chose $b = 1.0$, so the non-dimensional wavenumbers bS are the same as shown. The asymptotic formula is surprisingly accurate except for the two lowest attenuating wavenumbers. The wavenumber S_*° can be calculated by using low volume fraction expansions [11].

For *multipole scatterers*, where both n and m could potentially range from $-\infty$ to ∞ in (3.15), we can also calculate an infinite sequence of effective wavenumbers. To show this explicitly, we consider the limit of large bk , with $|k| \sim |S|$. In the opposite limit $bk \ll 1$, the Rayleigh limit, only one effective wavenumber is required [44,45].

At leading order, the asymptotic solution of (A 11) leads to the effective wavenumbers:

$$bS_p^{k\pm} = \sigma_p^{\pm} + i \log \left(\frac{|\sigma_p^{\pm} - a| \sqrt{a|\sigma_p^{\pm}|}}{r_c} \right), \quad (5.5)$$

$$\begin{cases} \sigma_p^+ = \theta_c + a + 2\pi p & \text{for } p > \left\lceil -\frac{\theta_c + a}{2\pi} \right\rceil, \\ \sigma_p^- = \theta_c + a - \frac{3\pi}{2} - 2\pi p & \text{for } p > \left\lceil \frac{\theta_c + a}{2\pi} - \frac{3}{4} \right\rceil, \end{cases} \quad (5.6)$$

$$r_c e^{i\theta_c} = -2inb^2 \sum_{n=-\infty}^{\infty} T_n, \quad r_c > 0 \quad \text{and} \quad -\pi \leq \theta_c \leq \pi, \quad (5.7)$$

for integer p . This confirms that there are an infinite number of effective wavenumbers for large scatterers, i.e. $bk \gg 1$. The distribution of these wavenumbers is similar to the monopole wavenumbers shown in Figure 4.

These asymptotic formulas (5.3) and (5.5) demonstrate the existence of multiple effective waves in the limit of small (monopole and Dirichlet) scatterers (5.3) and large scatterers (5.5). However, neither of these formulas, nor the low volume fraction expansions of the wavenumber [11], are able to accurately estimate the low attenuating backward travelling effective wavenumber such as S_1 shown in Figure 3 (in this case not related to the $S_1^{\circ\pm}$ and $S_1^{k\pm}$

given above). There is currently no way to analytically estimate these types of wavenumbers, even though they are necessary to accurately calculate transmission due to their small attenuation. The only approach it seems is to numerically solve (3.30).

6. Numerical results

Here we present numerical results for monopole scatterers, as these have explicit expressions for reflection (4.5) and the transmitted wave (4.4) (or more accurately the average scattering coefficients). We compare our analytic solution with a classical method that assumes only one effective wavenumber [11,13], and the Matching Method [18], recently proposed by the authors. It should be noted that all of these approaches aim to solve the same equation (2.14).

Note that for monopole scatterers, using only one effective wavenumber s_1 can, in some cases, lead to accurate results. However, for multipole scatterers (a more common scenario practically) this is rarely the case because, as shown by Figure 3, there can be at least two effective wavenumbers with low attenuation, and therefore both are needed to obtain accurate results.

For the numerical examples we use the parameters

$$T_0 = -\frac{J_0(ka_o)}{H_0^{(1)}(ka_o)}, \quad b = 1.001, \quad a_o = 0.5, \quad \theta_{\text{inc}} = \frac{\pi}{4}, \quad \phi = 30\%, \quad (6.1)$$

which implies that the number fraction $n \approx 0.38$ per unit area. When we choose to fix the wavenumber, as we do for Figure 5 and Figure 6, we use $kb = 1.001$. This leads to a wavelength ($2\pi/k$) which is roughly six times larger than the particle diameter. If the particle was, say, more than a hundred times smaller than the wavelength, then only one effective wavenumber in the sum (4.4) would be necessary to accurately calculate $A_0(X)$.

Figure 5. Compares the absolute value of the average field $A_0(x)$ calculated by different methods. The field $A_0(x)$ is closely related to the average transmitted wave [13]. The non-dimensional wavenumber $kb = 1.001$, the other parameters are given by (6.1), with their definitions explained by (2.4–2.9). Using the Wiener-Hopf solution (4.4), we approximate $A_0(x)$ by using either 352 effective wavenumbers s_1, s_2, \dots, s_{352} , or just 1 effective wavenumber s_1 . The Matching Method also accounts for multiple effective wavenumbers, and is described in [18]. The low volume fraction method assumes a low volume fraction expansion for just one effective wavenumber [11]. The small graph on the right is a magnification of the region around $x = 0$. Close to the boundary $x = 0$, both $A_0^1 e^{is_1 x}$ and the low volume fraction method are inaccurate, which would potentially lead to inaccurate predictions for transmission and reflection.

To start we compare the average scattering coefficient $A_0(x)$ calculated by the Wiener-Hopf solution (4.4) with other methods in Figure 5. The most accurate of these other methods is the Matching Method [18,46], and it closely agrees with the Wiener-Hopf solution when using 352 effective wavenumbers. The exception is the region close to the boundary $x = 0$, where the Wiener-Hopf solution experiences a rapid transition. The low volume fraction method is the most commonly used in the literature: it assumes a small particle volume fraction¹⁰ and just one effective wavenumber [11,13]. One significant conclusion we can draw from Figure 5 is that both the low volume fraction method and $A_0^1 e^{is_1 x}$ are inaccurate near the boundary $x = 0$. This means that both of these methods lead to inaccurate reflection coefficients.

In general, the Wiener-Hopf method does not lead to an explicit formula for the reflection coefficient (3.35), because we do not have an exact factorisation (3.23) for any truncated square matrices. However, there are methods [13,18,20,29,30,34] to calculate $A_n(x)$, from which we can obtain the reflection coefficient (2.12). The method [18] also accounts for multiple effective wavenumbers. So one important question is: when using (2.12), how many effective wavenumbers do we need to obtain an accurate reflection coefficient?

Figure 6. Demonstrates, with a log-log graph, how increasing the number of effective waves P leads to a more accurate reflection coefficient R^P , when using (4.6). The non-dimensional wavenumber is $kb = 1.001$, and the other parameters used are given by (6.1), with their definitions explained by (2.4–2.9). Here R is the reflection coefficient given by (4.5). The error $|R^P - R|$ continuously drops as P increases because of the rapid transition that occurs to $A_0(x)$ near the boundary $x = 0$, see Figure 5. However, methods such as the Matching Method [18] are able to accurately calculate the reflection coefficient without taking into account this rapid transition.

In Figure 6 we show how increasing the number of effective waves P reduces the error between R^P (4.6) and R (4.5). To calculate a highly accurate reflection coefficient R , we could use either (4.5) or the Matching Method [18,46], as both give approximately the same R .

Now we ask: how does the reflection coefficient (4.6), deduced via the Wiener-Hopf technique, compare with other methods across a broader range of wavenumbers. The result is shown in Figure 7, where R^O is a low volume fraction expansion¹¹ of just one effective wavenumber [13]. The reflection coefficient R^M is calculated from the Matching Method [18,46]. The general trend is clear: R^O becomes more inaccurate as we increase the background wavenumber kb . On the other hand both R^M and R agree closely over all k .

One result to note is the “instability” exhibited by the Wiener-Hopf solution near the boundary $x = 0$, see Figure 5. This instability occurs because we represented $A_0(x)$ as a superposition of truncated waves, which is only accurate as long as the discarded terms are small. So, for

¹⁰For the low volume fraction method we used a small volume fraction expansion for the wavenumber, but we numerically evaluated the wave amplitude. This is because the alternative, a small volume fraction expansion of the wave amplitude, led to poor results.

¹¹We use the reflection coefficient [13, equation (39)], rather than the explicit low volume fraction expansion [13, equation (40–41)]. This is because using equations (40–41) led to roughly double the error we show.

Figure 7. Compares different methods for calculating the reflection coefficient when varying the non-dimensional wavenumber kb . The other parameters used are given by (6.1), with their definitions explained in (2.4–2.9). Here \mathfrak{R} is given by the Wiener-Hopf solution (4.5), \mathfrak{R}^O uses a low volume fraction expansion of just one effective wavenumber [13], and \mathfrak{R}^M is calculated from the Matching Method [18].

a truncation number P , we can expect the instability to occur when $e^{i s_P x}$ is not small, i.e. $x \approx 1/\text{Im } s_P$. However, this instability does not affect the accuracy of the reflection coefficient (4.5) deduced by the Wiener-Hopf technique, as demonstrated by close agreement with the Matching Method in Figure 7.

7. Conclusion and Next Steps

The major result of this paper is to prove that the ensemble-averaged field in random particulate materials consists of a superposition of waves, with complex effective wavenumbers, for one fixed incident wavenumber. These effective wavenumbers are governed by the dispersion equation (5.1) and are independent of the angle of incidence θ_{inc} . We showed asymptotically in section 5 that this has an infinite number of solutions, and hence there are an infinite number of effective wavenumbers. The Wiener-Hopf technique also provides a simple and elegant expression for the reflection coefficient (3.35), whose form can be used to guide and assess methods to characterise microstructure [47,48].

To numerically implement the Wiener-Hopf technique, we considered particles that scatter only in their monopole mode in section 6. There we saw that when close to the interface of the half-space, a large number of effective wavenumbers were necessary to reach accurate agreement with an alternative method from the literature, the Matching Method as introduced by the authors in [18]. To obtain a constructive method via the Wiener-Hopf technique for general scatterers, and not just monopole scatterers, will require the factorisation of a matrix-function [31], which is challenging. For these reasons the Matching Method [18] is presently more effective than using the Wiener-Hopf technique. However, there is ongoing work to use approximate methods [33,34,49] which exploit the symmetry and properties of the matrix (3.15).

Moving forwards, this paper together with [18], establish accurate and robust solutions to the governing equation (2.14). These same methods can now be translated to three spatial dimensions and vectorial waves (e.g. elasticity and electromagnetics), with much of the groundwork already available [12,16,40]. Some clear challenges, that can now be addressed, are to verify the accuracy of the statistical assumptions used to deduce (2.14). These include the hole-correction and the quasicrystalline approximations. As these are now the only assumptions used, we could compare the solution of (2.14) with multipole methods [50,51] in order to investigate their accuracy and limits of validity.

A. The Fourier transformed kernel $\hat{\psi}_n(s)$

Here we calculate the Fourier transform (3.3) of $\psi_n(X)$ (2.13). To do so, it is simpler to use

$$F_n(X, Y) = (-1)^n H_n^{(1)}(kR) e^{in\Theta}. \quad (\text{A } 1)$$

Note that both $F_n(X, Y)$ and $e^{i(sX+Yk \sin \theta_{\text{inc}})}$ satisfy wave equations, with

$$\nabla^2 F_n(X, Y) = -k^2 F_n(X, Y) \quad \text{and} \quad \nabla^2 e^{i(sX+Yk \sin \theta_{\text{inc}})} = -S^2 e^{i(sX+Yk \sin \theta_{\text{inc}})},$$

where we used (2.13)₂ for the first equation and (3.19) for the second equation. This means that we can use Green's second identity to obtain

$$\begin{aligned} (k^2 - S^2) \int_{\mathcal{B}} e^{i(sX+Yk \sin \theta_{\text{inc}})} F_n(X, Y) dX dY \\ = \int_{\partial \mathcal{B}} \left[\frac{\partial e^{i(sX+Yk \sin \theta_{\text{inc}})}}{\partial \mathbf{n}} F_n(X, Y) - e^{i(sX+Yk \sin \theta_{\text{inc}})} \frac{\partial F_n(X, Y)}{\partial \mathbf{n}} \right] dz, \quad (\text{A } 2) \end{aligned}$$

for any area \mathcal{B} in which the integrand is analytic, where \mathbf{n} is the outwards pointing unit normal and dz is a differential length along the boundary $\partial \mathcal{B}$. To calculate $\hat{\psi}_n(s)$, we take the region \mathcal{B} to be defined by $R \geq b$, with (R, Θ) being the polar coordinates of (X, Y) , in which case the integral over \mathcal{B} converges because as $R \rightarrow \infty$ we have that

$$\begin{aligned} |e^{i(sX+Yk \sin \theta_{\text{inc}})} F_n(X, Y)| &\sim \frac{|e^{isR \cos \Theta} e^{ikR(1+\sin \Theta \sin \theta_{\text{inc}})}|}{\sqrt{\pi|k|R/2}} \\ &\leq \frac{|e^{-R(\text{Im } k(1-|\sin \theta_{\text{inc}}|) - |\text{Im } (s)|)}|}{\sqrt{\pi|k|R/2}} \rightarrow 0, \quad (\text{A } 3) \end{aligned}$$

exponentially fast when $|\text{Im } (s)| < \text{Im } k(1 - |\sin \theta_{\text{inc}}|)$. Under this restriction, and by assuming $S \neq \pm k$, (A 2) then leads to

$$\hat{\psi}_n(s) = \int_{R \geq b} e^{isX+ikY \sin \theta_{\text{inc}}} F_n(X, Y) dX dY = \frac{\mathcal{I}_n(b)}{k^2 - S^2}, \quad (\text{A } 4)$$

by using $sX + Yk \sin \theta_{\text{inc}} = RS \cos(\theta - \theta_S)$ from (3.19) and

$$\begin{aligned} \mathcal{I}_n(R) &= \int_0^{2\pi} -\frac{\partial e^{iSR \cos(\Theta - \theta_S)}}{\partial R} F_n(k\mathbf{X}) + e^{iSR \cos(\Theta - \theta_S)} \frac{\partial F_n(k\mathbf{X})}{\partial R} R d\Theta \\ &= (-1)^n \int_0^{2\pi} e^{iSR \cos(\Theta - \theta_S)} e^{in\Theta} [kH_n^{(1)'}(kR) - iS \cos(\Theta - \theta_S) H_n^{(1)}(kR)] R d\Theta \\ &= (-1)^n \int_0^{2\pi} \sum_{m=-\infty}^{\infty} i^m J_m(SR) e^{im(\Theta - \theta_S)} \left[k e^{in\Theta} H_n^{(1)'}(kR) \right. \\ &\quad \left. - \frac{iS}{2} (e^{i(n+1)\Theta - i\theta_S} + e^{i(n-1)\Theta + i\theta_S}) H_n^{(1)}(kR) \right] R d\Theta \\ &= 2\pi (-i)^n R e^{in\theta_S} \left[k J_n(SR) H_n^{(1)'}(kR) - S J_n'(SR) H_n^{(1)}(kR) \right], \quad (\text{A } 5) \end{aligned}$$

where J_n is the Bessel function of the first kind, and we used the Jacobi-Anger expansion on $e^{iSR \cos(\Theta - \theta_S)}$, integrated over Θ and used the identity $J_{n-1}(SR) - J_{n+1}(SR) = 2J_n'(SR)$. In summary

$$\hat{\psi}_n(s) = 2\pi \frac{(-i)^n e^{in\theta_S}}{\alpha^2 - s^2} N_n(bS), \quad (\text{A } 6)$$

when the condition (3.10) is satisfied, with N_n given by (3.17).

Below we establish some useful properties for $\hat{\psi}_n(s)$. In particular, we show that $\hat{\psi}_n(s)$ has no branch-points.

The function $N_n(bS)$, for integer values of n , can be expanded around $S = 0$ as

$$N_n(bS) = S^{|n|} \sum_{m=0}^{\infty} c_{m|n|} S^{2m}, \quad (\text{A } 7)$$

where the $c_{m|n|}$ are some constants that depend on m and $|n|$, and the radius of convergence of the series above is infinite. Using (3.19) we can write

$$e^{in\theta_S} = e^{i \operatorname{sgn}(n)|n|\theta_S} = (\cos \theta_S + \operatorname{sgn}(n)i \sin \theta_S)^{|n|} = (s + \operatorname{sgn}(n)ik \sin \theta_{\text{inc}})^{|n|} S^{-|n|}. \quad (\text{A } 8)$$

Substituting (A 7) and (A 8) in (A 6) results in

$$\hat{\psi}_n(s) = \frac{2\pi(-i)^n}{\alpha^2 - s^2} (s + \operatorname{sgn}(n)ik \sin \theta_{\text{inc}})^{|n|} \sum_{m=0}^{\infty} c_{m|n|} S^{2m}. \quad (\text{A } 9)$$

Because $S^2 = s^2 + k^2 \sin^2 \theta_{\text{inc}}$, we can immediately see from the above that $\hat{\psi}_n(s)$ has no branch-points. Additionally we can establish the properties:

$$\hat{\psi}_n(s) = \hat{\psi}_{-n}(-s) = \hat{\psi}_n(-s) e^{2in\theta_S} (-1)^n. \quad (\text{A } 10)$$

B. Asymptotic location of the wavenumbers

Here we explicitly calculate a sequence of effective wavenumbers S_p , assuming $|S_p|$ large and increasing with p , and $\operatorname{Im} S_p > 0$, that asymptotically satisfy (5.1). A key step is to approximate the terms appearing in (3.16), such as

$$J_n(bS) \sim \frac{e^{\frac{i\pi}{4} + \frac{in\pi}{2} - ibS}}{\sqrt{2\pi bS}} \quad \text{and} \quad J'_n(bS) \sim \frac{e^{-\frac{i\pi}{4} + \frac{in\pi}{2} - ibS}}{\sqrt{2\pi bS}}, \quad (\text{A } 1)$$

for large $|bS|$, where the terms e^{ibS} are discarded as $\operatorname{Im} bS \rightarrow \infty$.

Monopole scatterers The simplest case is for monopole scatterers, where $n = m = 0$ in (3.16), and the effective wavenumber S satisfies

$$b^2 \det \mathbf{G} = (bS)^2 - (bk)^2 + 2\pi n b^2 T_0 N_0(bS) \sim (bS)^2 - c\sqrt{bS} e^{-ibS} = 0, \quad (\text{A } 2)$$

where $c = \sqrt{2\pi n} b^2 T_0 H_0^{(1)}(kb) e^{-\frac{i\pi}{4}}$. Here we used (A 1), and ignored terms which are algebraically smaller than bS . To find the root of the above we substitute

$$bS = x + i \log y, \quad (\text{A } 3)$$

where x and y are real, and $|x|$ and y are large with $y > 1$. This leads to

$$(x + i \log y)^{3/2} - c e^{-ix} y = 0. \quad (\text{A } 4)$$

For the logarithm and square root we use the typical branch cut $(-\infty, 0)$ and take positive values of the functions for positive arguments. For the above to be satisfied to leading order then $x^{3/2} \sim y$, which reduces the above equation to

$$x^{3/2} \sim r_c e^{i(\theta_c - x)} y, \quad (\text{A } 5)$$

where we substituted $c = r_c e^{i\theta_c}$, for real scalars r_c and θ_c . Equating the real and imaginary parts of the above leads to

$$x \sim \theta_c + 2\pi p \quad \text{and} \quad y \sim \frac{1}{r_c} (\theta_c + 2\pi p)^{3/2} \quad \text{for } p > -\frac{\theta_c}{2\pi} \quad (\text{A } 6)$$

$$x \sim \theta_c - \frac{3\pi}{2} + 2\pi p \quad \text{and} \quad y \sim \frac{1}{r_c} (-\theta_c + \frac{3\pi}{2} - 2\pi p)^{3/2} \quad \text{for } p < \frac{3}{4} - \frac{\theta_c}{2\pi}, \quad (\text{A } 7)$$

for integers p . From this we can identify that, at leading order, the effective wavenumbers are given by (5.3).

Multipole scatterers: With the same method used above, we can also demonstrate the existence of multiple effective wavenumbers for $n, m = -M, -M + 1, \dots, M$ in (3.16). To show this explicitly, we consider bk to be the same order as bS , that is $|k| \sim |S|$.

By considering bk large, we can approximate

$$H_n^{(1)}(bk) \sim e^{i(bk - \frac{\pi}{4} - \frac{n\pi}{2})} \sqrt{\frac{2}{\pi bk}} \quad \text{and} \quad H_n^{(1)'}(bk) \sim e^{i(bk + \frac{\pi}{4} - \frac{n\pi}{2})} \sqrt{\frac{2}{\pi bk}}, \quad (\text{A } 8)$$

combining this with (A 1) and considering $|k| \sim |S|$, the term (3.16) at leading order becomes

$$b^2 G_{mn} = d_0 \delta_{mn} + c_0 T_m, \quad (\text{A } 9)$$

where

$$d_0 = (bS)^2 - (bk)^2, \quad \text{and} \quad c_0 = 2nb^2 \frac{i(k+S)}{\sqrt{kS}} e^{ib(k-S)}.$$

By simple rearrangement of the determinant we find that¹²

$$\det(b^2 \mathbf{G}) = d_0^{2M} \left(d_0 + c_0 \sum_{m=-M}^M T_m \right). \quad (\text{A } 10)$$

Note that $d_0 \neq 0$, i.e. $S \neq \pm k$, was necessary to reach the condition (3.10), which was used to calculate the Fourier transforms (3.13). Taking this into consideration, and taking the limit $M \rightarrow \infty$, the effective wavenumbers S must satisfy

$$d_0 + c_0 \sum_{m=-M}^M T_m = 0 \implies bS - bk = -2nb^2 \sum_{m=-\infty}^{\infty} T_m \frac{e^{ib(k-S)}}{b\sqrt{kS}}. \quad (\text{A } 11)$$

Using an asymptotic expansion analogous to (A 3), the above leads to the effective wavenumbers (5.5).

C. Equivalent determinants

For any square matrices \mathbf{A} and \mathbf{B} , and scalar c , if $A_{nm} = B_{nm}c^{n-m}$ (not employing the summation convention), then

$$\det \mathbf{A} = \det \mathbf{B}. \quad (\text{A } 1)$$

This follows simply by defining the diagonal matrix $C_{nm} = \delta_{nm}c^n$, which leads to $\mathbf{A} = \mathbf{C}\mathbf{B}\mathbf{C}^{-1}$, and $\det(\mathbf{C}\mathbf{B}\mathbf{C}^{-1}) = \det \mathbf{C} \det \mathbf{B} \det \mathbf{C}^{-1} = \det \mathbf{B}$.

Data Accessibility. We provide the code to generate all the graphs in [46].

Authors' Contributions. I.D.A and A.L.G. conceived of the study. A.L.G. drafted the manuscript. A.L.G., I.D.A, and W.J.P edited the manuscript and gave final approval for publication.

Competing Interests. We have no competing interests.

Funding. This work was funded by EPSRC (EP/M026205/1, EP/L018039/1, EP/S019804/1) including via the Isaac Newton Institute (EP/K032208/1, EP/R014604/1) and partial support from the UKAN grant EPSRC (EP/R005001/1).

References

1. Pinfield VJ, Challis RE. 2013 Emergence of the coherent reflected field for a single realisation of spherical scatterer locations in a solid matrix. *Journal of Physics: Conference Series* **457**, 012009.
2. Mishchenko MI, Travis LD, Lacis AA. 2006 *Multiple Scattering of Light by Particles: Radiative Transfer and Coherent Backscattering*. Cambridge University Press.

¹²The determinant of $b^2 \mathbf{G}$ equals the product of its eigenvalues. The eigenvector $(T_{-M}, \dots, T_M)^T$ gives the eigenvalue $d_0 + c_0 \sum_m T_m$, while all other eigenvalues equal d_0 .

3. Mishchenko MI, Dlugach JM, Yurkin MA, Bi L, Cairns B, Liu L, Panetta RL, Travis LD, Yang P, Zakharova NT. 2016 First-principles modeling of electromagnetic scattering by discrete and discretely heterogeneous random media. *Physics Reports* **632**, 1–75. arXiv: 1605.06452.
4. Tsang L, Ishimaru A. 1987 Radiative Wave Equations for Vector Electromagnetic Propagation in Dense Nontenuous Media. *Journal of Electromagnetic Waves and Applications* **1**, 59–72.
5. Tsang L, Chen CT, Chang ATC, Guo J, Ding KH. 2000 Dense media radiative transfer theory based on quasicrystalline approximation with applications to passive microwave remote sensing of snow. *Radio Science* **35**, 731–749.
6. Foldy LL. 1945 The multiple scattering of waves. I. General theory of isotropic scattering by randomly distributed scatterers. *Physical Review* **67**, 107.
7. Lax M. 1951 Multiple Scattering of Waves. *Reviews of Modern Physics* **23**, 287–310.
8. Fikioris JG, Waterman PC. 1964 Multiple Scattering of Waves. II. “Hole Corrections” in the Scalar Case. *Journal of Mathematical Physics* **5**, 1413–1420.
9. Tsang L, Kong JA. 1983 Scattering of electromagnetic waves from a half space of densely distributed dielectric scatterers. *Radio Science* **18**, 1260–1272.
10. Tishkovets VP, Petrova EV, Mishchenko MI. 2011 Scattering of electromagnetic waves by ensembles of particles and discrete random media. *Journal of Quantitative Spectroscopy and Radiative Transfer* **112**, 2095–2127.
11. Linton CM, Martin PA. 2005 Multiple scattering by random configurations of circular cylinders: Second-order corrections for the effective wavenumber. *The Journal of the Acoustical Society of America* **117**, 3413.
12. Linton CM, Martin PA. 2006 Multiple Scattering by Multiple Spheres: A New Proof of the Lloyd–Berry Formula for the Effective Wavenumber. *SIAM Journal on Applied Mathematics* **66**, 1649–1668.
13. Martin PA. 2011 Multiple scattering by random configurations of circular cylinders: Reflection, transmission, and effective interface conditions. *The Journal of the Acoustical Society of America* **129**, 1685–1695.
14. Norris AN, Conoir JM. 2011 Multiple scattering by cylinders immersed in fluid: High order approximations for the effective wavenumbers. *The Journal of the Acoustical Society of America* **129**, 104–113.
15. Gower AL, Smith MJA, Parnell WJ, Abrahams ID. 2018 Reflection from a multi-species material and its transmitted effective wavenumber. *Proc. R. Soc. A* **474**, 20170864.
16. Conoir JM, Norris AN. 2010 Effective wavenumbers and reflection coefficients for an elastic medium containing random configurations of cylindrical scatterers. *Wave Motion* **47**, 183–197.
17. Norris AN, Luppé F, Conoir JM. 2012 Effective wave numbers for thermo-viscoelastic media containing random configurations of spherical scatterers. *The Journal of the Acoustical Society of America* **131**, 1113–1120.
18. Gower AL, Parnell WJ, Abrahams ID. 2018 Multiple Waves Propagate in Random Particulate Materials. *arXiv:1810.10816 [physics]*. arXiv: 1810.10816.
19. Crighton DG, Dowling A, Ffowcs Williams J, Heckl M, Leppington F. 1992 *Modern Methods in Analytical Acoustics*. Springer-Verlag.
20. Lawrie JB, Abrahams ID. 2007 A brief historical perspective of the Wiener–Hopf technique. *Journal of Engineering Mathematics* **59**, 351–358.
21. Noble B. 1988 *Methods Based on the Wiener-Hopf Technique*. vol. 67. American Mathematical Society 2nd unexpurgated edition edition.
22. Martin PA, Abrahams ID, Parnell WJ. 2015 One-dimensional reflection by a semi-infinite periodic row of scatterers. *Wave Motion* **58**, 1–12.
23. Norris A, Wickham GR. 1995 Acoustic diffraction from the junction of two flat plates. *Proceedings of the Royal Society of London. Series A: Mathematical and Physical Sciences* **451**, 631–655.
24. Haslinger SG, Movchan NV, Movchan AB, Jones IS, Craster RV. 2016 Controlling flexural waves in semi-infinite platonic crystals. *arXiv:1609.02787 [physics]*. arXiv: 1609.02787.
25. Tymis N, Thompson I. 2014 Scattering by a semi-infinite lattice and the excitation of Bloch waves. *The Quarterly Journal of Mechanics and Applied Mathematics* **67**, 469–503.
26. Haslinger SG, Jones IS, Movchan NV, Movchan AB. 2017 Semi-infinite herringbone waveguides in elastic plates. *arXiv:1712.01827 [physics]*. arXiv: 1712.01827.
27. Albani M, Capolino F. 2011 Wave dynamics by a plane wave on a half-space metamaterial made of plasmonic nanospheres: a discrete Wiener–Hopf formulation. *JOSA B* **28**, 2174–2185.

28. Abrahams ID. 1997 On the Solution of Wiener–Hopf Problems Involving Noncommutative Matrix Kernel Decompositions. *SIAM Journal on Applied Mathematics* **57**, 541–567.
29. Abrahams ID. 1996 Radiation and scattering of waves on an elastic half-space; A non-commutative matrix Wiener-Hopf problem. *Journal of the Mechanics and Physics of Solids* **44**, 2125–2154.
30. Abrahams ID, Wickham GR. 1990 General Wiener–Hopf Factorization of Matrix Kernels with Exponential Phase Factors. *SIAM Journal on Applied Mathematics* **50**, 819–838.
31. Rogosin S, Mishuris G. 2016 Constructive methods for factorization of matrix-functions. *IMA Journal of Applied Mathematics* **81**, 365–391.
32. Kisil A. 2018 An Iterative Wiener–Hopf Method for Triangular Matrix Functions with Exponential Factors. *SIAM Journal on Applied Mathematics* **78**, 45–62.
33. Abrahams ID. 2000 The application of Padé approximants to Wiener-Hopf factorization. *IMA Journal of Applied Mathematics* **65**, 257–281.
34. Abrahams ID. 1987 Scattering of sound by two parallel semi-infinite screens. *Wave Motion* **9**, 289–300.
35. Ganesh M, Hawkins SC. 2010 A far-field based T-matrix method for two dimensional obstacle scattering. *ANZIAM Journal* **51**, 215–230.
36. Ganesh M, Hawkins SC. 2017 Algorithm 975: TMATROM—A T-Matrix Reduced Order Model Software. *ACM Trans. Math. Softw.* **44**, 9:1–9:18.
37. Mishchenko MI. 1991 Light scattering by randomly oriented axially symmetric particles. *JOSA A* **8**, 871–882.
38. Mishchenko MI. 1993 Light scattering by size–shape distributions of randomly oriented axially symmetric particles of a size comparable to a wavelength. *Applied Optics* **32**, 4652–4666.
39. Waterman PC. 1971 Symmetry, Unitarity, and Geometry in Electromagnetic Scattering. *Physical Review D* **3**, 825–839.
40. Kristensson G. 2015 Coherent scattering by a collection of randomly located obstacles – An alternative integral equation formulation. *Journal of Quantitative Spectroscopy and Radiative Transfer* **164**, 97–108.
41. Bleistein N, Handelsman RA. 1986 *Asymptotic expansions of integrals*. Courier Corporation.
42. Gokhberg IC, Krein MG. 1960 Systems of integral equations on the half-line with kernels depending on the difference of the arguments. **14**, 217 – 287.
43. Martin PA, Maurel A. 2008 Multiple scattering by random configurations of circular cylinders: Weak scattering without closure assumptions. *Wave Motion* **45**, 865–880.
44. Parnell WJ, Abrahams ID. 2010 Multiple point scattering to determine the effective wavenumber and effective material properties of an inhomogeneous slab. *Waves in Random and Complex Media* **20**, 678–701.
45. Parnell WJ, Abrahams ID, Brazier-Smith PR. 2010 Effective Properties of a Composite Half-Space: Exploring the Relationship Between Homogenization and Multiple-Scattering Theories. *The Quarterly Journal of Mechanics and Applied Mathematics* **63**, 145–175.
46. Gower AL. EffectiveWaves.jl: A package to calculate ensemble averaged waves in heterogeneous materials. <https://github.com/arturgower/EffectiveWaves.jl/tree/v0.2.1>.
47. Roncen R, Fella ZEA, Simon F, Piot E, Fella M, Ogam E, Depollier C. 2018 Bayesian inference for the ultrasonic characterization of rigid porous materials using reflected waves by the first interface. *The Journal of the Acoustical Society of America* **144**, 210–221.
48. Gower AL, Gower RM, Deakin J, Parnell WJ, Abrahams ID. 2018 Characterising particulate random media from near-surface backscattering: A machine learning approach to predict particle size and concentration. *EPL (Europhysics Letters)* **122**, 54001.
49. Veitch BH, Abrahams ID. 2007 On the commutative factorization of nxn matrix Wiener–Hopf kernels with distinct eigenvalues. *Proceedings of the Royal Society A: Mathematical, Physical and Engineering Sciences* **463**, 613–639.
50. Martin PA. 2006 *Multiple Scattering: Interaction of Time-Harmonic Waves with N Obstacles*. Cambridge: Cambridge University Press.
51. Gower AL, Deakin J. 2019 MultipleScattering.jl: A Julia library for simulating, processing, and plotting multiple scattering of acoustic waves. <https://github.com/jondea/MultipleScattering.jl>.

Appendix B

A Proof that Multiple Waves Propagate in Ensemble-Averaged Particulate Materials: response to referees

August 2, 2019

We are grateful for the very positive comments from the referees and Board Member and thank them for all of their input and suggestions to improve the paper. Below we have responded to referee questions, indicating where we have made modifications to the manuscript. For minor points, we have just made the suggested changes, without mentioning them below.

All changes made to this paper, except minor changes to the text, are marked in blue in a highlighted version of the paper: `gower-2019-Highlighted.pdf`. We have also provided a non-highlighted version of the revised article.

Referee: 1

Main comments

1. The assumption that the free-space (vacuum) wavenumber has a small imaginary part seems critical to all of the derived results, e.g., see eq. (3.10). Physically, k_0 (wavenumber inside a particle) may and should contain a small imaginary part but this is not the case for the free-space. I understand the mathematical necessity to add a small imaginary part to k but I wonder if one could obtain similar results for strictly real k . I suspect the answer is no and thus, the relevance of this manuscript to real physical systems is in question. It would be great if authors could elaborate on this point beyond "[The] dissipation will facilitate the use of the Wiener-Hopf technique, and after reaching the solution we can take k to be real."

We agree with the referee that the relevance to real physical systems is indeed very important. This is why our approach is valid for strictly real incident wavenumbers k as well as for complex wavenumbers. For halfspace problems, the analysis/results for $\text{Im } k < 0$ (in our case not physically viable) are very different in comparison to $\text{Im } k > 0$. For this reason, we have to take care to define the correct physical behaviour for $\text{Im } k = 0$. By taking $\text{Im } k = \epsilon > 0$, this not only greatly facilitates the analysis, it also allows us to recover the correct physical results for $\text{Im } k = 0$. To do so, we need only take the limit $\epsilon \rightarrow 0$ after obtaining the solution, then, if a solution exists for $\text{Im } k = 0$, this recovers the physically correct limit; such a limiting process has been attributed to Lighthill. There are other methods which avoid the need to introduce this dissipation. One such way would be to augment our function space and consider distributions, so we could then make sense of the integral (3.9) for $\text{Im } k = 0$. However, we feel this would add too much notation for little gain. We have added a footnote addressing this point just below equation (2.5).

2. I suspect that if k_0 has even a small imaginary component, then one will get only one dominant effective wavenumber, see fig. 3. Is this true?

We do not find this to be the case. The dispersion equation (5.1) is a continuous function of the material properties, with k_0 included. So a small change in $\text{Im } k_0$ leads to a consequent small change in the effective wavenumbers. To verify, we conducted a parameter search for $\text{Im } k_0 > 0$ near the case shown in Figure 3. We found that having two dominate effective wavenumbers was a very stable phenomenon, and always appeared. However, it is true to say that in general we do not have much intuition about how the imaginary part of k_0 affects these effective wavenumbers.

3. Please state that you are using $\exp(-i\omega t)$ time factor. (This choice is consistent with the Hankel function of the 1st kind in eq. (2.10)).

Done.

Minor typo 4. Eq. (3.1): should be $x_1 > 0$ and $x_1 \leq 0$

We have now changed equation (2.14) to be valid for $x_1 \geq 0$, and have therefore kept the cases $x_1 \geq 0$ and $x_1 < 0$ in equation (3.1), so as to be consistent.

Referee: 2

1. The abstract should more explicitly state the main assumptions, boundary conditions and conclusions of the paper. In particular the influence of monopole/multipole scatters (see page 15, lines 8-12) should be stated. 2. The Matching Method should be briefly mentioned in the abstract as it forms the basis for the work presented.

We have now rewritten the abstract to accommodate these points, though are limited due to space; i.e. to state the problem and conclusions more clearly, mentioned the matching method, and that the reflection coefficient, deduced by the Wiener-Hopf technique, can be explicitly evaluated for monopole scatterers.

3. Page 4. The spatial extent of the incident plane wave is not mentioned as an assumption that simplifies the analysis. A plane wave with spatial extent in y would encounter different boundary conditions along the boundary as the relative acoustic impedance changes, due to local changes in the particulate material. Also near and far field effects would need to be considered in the analysis.

We agree that having an incident plane wave of the form $e^{i(\alpha x + \beta y)}$ for $x < 0$ and $y \in \mathbb{R}$ does indeed simplify the analysis as long as the statistics of the distribution of particles is invariant in the orthogonal y -direction. Clearly there will be local changes along the boundary for any specific configuration of particles, but the act of ensemble averaging removes these local changes, and leads to the field $\langle u(x, y) \rangle$ having a translational symmetry along y , see [1] for details. We have added a footnote just below equation (2.10) on this.

We are not sure which specific near and far field effects the referee refers to. There are a few different fields which could be considered near-fields: when two particles are close to each other we could describe their near field effects, which are included in the governing equation (2.14). We could also consider the fields near the boundary $x = 0$ to be near-fields, which is where these multiple effective waves have the greatest affect. These types of near-field effects, and fields far from the boundary, are all considered in our analysis. Only if the region containing the inclusions was finite in y would we need to consider global far-field effects in our work (i.e. far from the multiple scattering region); but as already mentioned, we preclude this in our analysis.

4. Page 4, lines 47-49. The statement about the agreement between the methods is not clearly referenced. It could be read as a conclusion of the present work.

Agreed, we have reworded.

5. Page 5, line 43. The numerical order of the referencing is not correct.

We assume the referee is referring to Page 6, line 43, where we have now corrected the order of the referenced equations.

6. Page 13, lines 15-16. The conclusion that the wavenumbers S_p are independent of angle of incident should be mentioned in the conclusions section.

Done.

7. Page 18, lines 15-25. Check sentence..

We have reworded this sentence.

References

- [1] A. L. GOWER, M. J. A. SMITH, W. J. PARNELL, AND I. D. ABRAHAMS, *Reflection from a multi-species material and its transmitted effective wavenumber*, Proc. R. Soc. A, 474 (2018), p. 20170864, <https://doi.org/10.1098/rspa.2017.0864>, <http://rspa.royalsocietypublishing.org/content/474/2212/20170864> (accessed 2018-04-22).